# Generation is Required for Data-Efficient Perception

## Abstract

Visual perception in the human brain is often thought to result from inverting a generative *decoder* that maps latents to images. In contrast, today's most successful vision models are non-generative, relying on an *encoder* that maps images to latents without inverting an image decoder. This raises the question of whether generation is required for machines to achieve human-level visual perception. In this work, we approach this question from the perspective of data efficiency, a core feature of human perception. Specifically, we investigate whether *compositional generalization* is achievable, both in theory and practice, using generative and non-generative methods. We first formalize the inductive biases required to guarantee compositional generalization in generative (decoder-based) and non-generative (encoder-based) methods. We then provide theoretical results suggesting that such inductive biases cannot be enforced on an encoder through practical means such as regularization or architectural constraints. In contrast, we show that enforcing the inductive biases on a decoder is straightforward, enabling compositional generalization through inverting the decoder. We highlight how this inversion can be performed efficiently, either online through gradient-based search or offline through generative replay. Empirically, we train a range of non-generative methods on photorealistic image datasets, finding they often fail to generalize compositionally and require large-scale pretraining to improve generalization. By comparison, generative methods yield significant improvements in compositional generalization, without requiring additional data, by leveraging suitable inductive biases on a decoder along with search and replay.

## 1 Introduction

Perceiving the visual world requires forming internal representations of sensory input. Two opposing views exist for how these representations should be acquired. The **generative view** posits that representations result from inverting an internal generative model, or *decoder*, to infer the latent variables that give rise to the visual input (Friston and Stephan, 2007; Hinton, 2007; Olshausen, 2014; von Helmholtz, 1867). Conversely, the **non-generative view** holds that representations result from a feedforward *encoder* that directly maps visual input to latent variables without inverting a decoder (Gibson, 1979; LeCun, 2022; Yamins et al., 2014). A core problem in AI is to understand which of these paradigms should be adopted to build machines with *human-level visual perception*.

In recent years, consensus around this problem has shifted, following breakthroughs in non-generative methods for representation learning (Caron et al., 2021; Oquab et al., 2024; Radford et al., 2021; Tschannen et al., 2025). These methods, trained with self- or weak supervision, now enable unprecedented performance on perceptual tasks such as object recognition (Siméoni et al., 2025) and image captioning (Beyer et al., 2024; Fan et al., 2025). This progress has given rise to a common assumption that non-generative methods provide the most promising path toward human-level visual perception, while generative approaches are not necessary (Balestriero and LeCun, 2024).

Yet, despite their remarkable performance, current non-generative methods fall short in another key pillar of human visual perception: *data efficiency*. Specifically, these methods rely on web-scale datasets in which different visual concepts are encountered across diverse contexts, with high frequency (Udandarao et al., 2024), and often with language supervision (Zhang et al., 2024). In

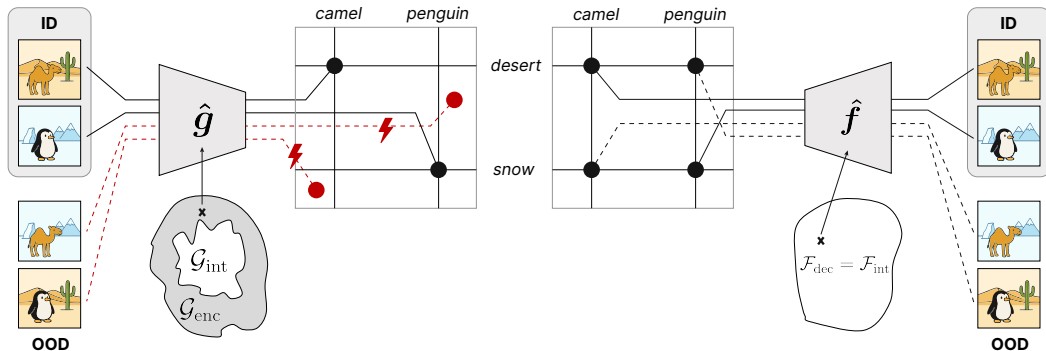

Figure 1: **Generative vs. non-generative compositional generalization.** We assume in-domain (ID) and out-of-domain (OOD) images arise from a latent variable model through an unknown generator $\boldsymbol{f} \in \mathcal{F}_{\text{int}}$, with inverse $\boldsymbol{g} \in \mathcal{G}_{\text{int}}$. Guaranteeing compositional generalization for a generative approach requires constraining a *decoder* such that $\hat{\boldsymbol{f}} \in \mathcal{F}_{\text{int}}$, and for a non-generative approach, an encoder such that $\hat{\boldsymbol{g}} \in \mathcal{G}_{\text{int}}$ (Sec. 2). We show theoretically in (Sec. 3) that placing such constraints on an encoder is generally infeasible with practical approaches while for a decoder it is straightforward. Empirically, this tends to manifest in an encoder yielding incorrect representations for OOD images (Sec. 5.2). In contrast, a decoder is able to correctly generate such images enabling compositional generalization through inversion (Sec. 4, 5.2).

contrast, human children achieve robust visual perception through much more constrained data, encountering concepts only a handful of times, mainly in the same settings (e.g., the home), and with little supervision (Lake et al., 2017; Tenenbaum et al., 2011). To reach this level of data efficiency, it has been conjectured across several disciplines (Kilbertus et al., 2018; Lake et al., 2015; Peters et al., 2024) that a generative approach may be necessary. This raises a key question: Can non-generative approaches to perception also achieve human-level data efficiency, or is generation required?

In this work, we approach this question by analyzing, both theoretically and empirically, whether generative and non-generative methods can achieve *compositional generalization* (Fodor and Pylyshyn, 1988; Greff et al., 2020). Compositional generalization is the ability to perceive out-of-domain (OOD) scenes containing unseen combinations of concepts, e.g., a dog in the park after only seeing dogs in the house. It is thus essential for realizing the data efficiency of human perception.

**Structure and Contributions.** We build upon Brady et al. (2025) to formalize the constraints required to guarantee compositional generalization for both generative (decoder-based) and non-generative (encoder-based) approaches. In Sec. 3, we show theoretically that enforcing such constraints on encoders is generally infeasible, as they depend on the geometry of out-of-domain regions of the data manifold, which is unknown. In contrast, we show that for generative models the constraints are not data-dependent and can be imposed directly through regularization or architectural design. These results suggest that inversion of a decoder is necessary to guarantee compositional generalization. In Sec. 4, we describe how such inversion can be implemented efficiently: in-distribution via an autoencoder, and out-of-distribution via gradient-based search (Sec. 4.1) and generative replay (Sec. 4.2). Finally, in Sec. 5, we empirically evaluate compositional generalization using photorealistic image data containing concepts such as animals and backgrounds (Bordes et al., 2023). We find that non-generative models frequently fail to generalize compositionally on this data, requiring large-scale pretraining to succeed (Sec. 5.2). In contrast, generative methods leveraging search and replay achieve substantial gains in OOD performance without requiring additional data.

## 2 PROBLEM SETUP

**Perception.** We begin by formalizing *visual perception*. To this end, we assume that images $\boldsymbol{x} \in \mathcal{X} \subset \mathbb{R}^{d_x}$ arise from a latent variable model. Specifically, we assume $\boldsymbol{x}$ is generated from a latent vector $\boldsymbol{z} \in \mathcal{Z} := \mathbb{R}^{d_z}$ by a diffeomorphic generator $\boldsymbol{f} : \mathcal{Z} \to \mathcal{X}$, i.e., $\boldsymbol{x} = \boldsymbol{f}(\boldsymbol{z})$. Visual concepts in $\boldsymbol{x}$ (e.g. "camel" and "desert" in Fig. 1) are modelled as $K$ distinct *slots* of latents $\boldsymbol{z}_k \in \mathbb{R}^m$ such that $\boldsymbol{z} = (\boldsymbol{z}_1, ..., \boldsymbol{z}_K)$ (Brady et al., 2025). Now, assume we have a representation of an image $\hat{\boldsymbol{z}} = \phi(\boldsymbol{x})$, where $\phi : \mathbb{R}^{d_x} \to \mathcal{Z}$. We define perception as the ability to invert the generator $\boldsymbol{f}$ via $\phi$ to recover the slots $\boldsymbol{z}_k$ that generated $\boldsymbol{x}$. In general, recovering $\boldsymbol{z}_k$ exactly is impossible. Thus,

we only require that $\phi$ inverts $\boldsymbol{f}$ up to re-parameterizations and permutations of the slots. Formally, let $\boldsymbol{h}_\pi$ be a function composed of slot-wise bijections $\boldsymbol{h}_k : \mathbb{R}^m \to \mathbb{R}^m$ and permutations $\pi$, i.e., $\boldsymbol{h}_\pi(\boldsymbol{z}) := \{\boldsymbol{h}_k(\boldsymbol{z}_{\pi(k)})\}_{k=1}^K$. Perception on a set $\mathcal{Z}^S \subseteq \mathcal{Z}$ requires that there exist an $\boldsymbol{h}_\pi$ such that

$$\forall \boldsymbol{z} \in \mathcal{Z}^S, \ \phi(\boldsymbol{f}(\boldsymbol{z})) = \boldsymbol{h}_\pi(\boldsymbol{z}). \tag{2.1}$$

Eq. (2.1) takes the perspective of perception as an inverse problem (Tenenbaum et al., 2011), but with respect to the ground-truth generator $\boldsymbol{f}$. This contrasts with a task-based view (Yamins and DiCarlo, 2016) where perception is defined with respect to solving a downstream task. We note that the task-based view can be framed as a special case of Eq. (2.1), by treating task-specific predictions as the latent variables to be recovered by $\phi$. Moreover, if a representation satisfying Eq. (2.1) is learned, downstream tasks such as object classification can be solved via a simple readout applied independently to each inferred slot $\hat{\boldsymbol{z}}_k$ (see Sec. 5).

**Generative and non-generative approaches.** Using Eq. (2.1), we now characterize the *generative* and *non-generative* approaches to perception. For the generative approach, representations are obtained by inverting a learned *decoder* $\hat{\boldsymbol{f}} : \mathcal{Z} \to \mathbb{R}^{d_x}$, i.e., $\phi(\boldsymbol{x}) = \hat{\boldsymbol{f}}^{-1}(\boldsymbol{x})$. For this to satisfy Eq. (2.1), the decoder $\hat{\boldsymbol{f}}$ must *identify* the ground-truth generator $\boldsymbol{f}$ such that for $\boldsymbol{z} \in \mathcal{Z}$

$$\hat{\boldsymbol{f}}(\boldsymbol{h}_\pi(\boldsymbol{z})) = \boldsymbol{f}(\boldsymbol{z}). \tag{2.2}$$

Alternatively, for the non-generative approach, a representation is defined as $\phi(\boldsymbol{x}) = \hat{\boldsymbol{g}}(\boldsymbol{x})$, where $\hat{\boldsymbol{g}} : \mathbb{R}^{d_x} \to \mathcal{Z}$ is a learned *encoder* which *is not* constructed to invert a decoder $\hat{\boldsymbol{f}}$. For this to satisfy Eq. (2.1), $\hat{\boldsymbol{g}}$ must identify the inverse generator $\boldsymbol{g} := \boldsymbol{f}^{-1}$ such that for $\boldsymbol{x} \in \mathcal{X}$

$$\hat{\boldsymbol{g}}(\boldsymbol{x}) = \boldsymbol{h}_\pi(\boldsymbol{g}(\boldsymbol{x})). \tag{2.3}$$

We emphasize that the difference between the generative and non-generative approaches is not whether an encoder or decoder is used. Instead, it is whether a representation satisfying Eq. (2.1) is obtained by learning an approximation $\hat{\boldsymbol{f}}$ of the ground-truth generator (Eq. (2.2)) and then inverting this model, or by learning an approximation $\hat{\boldsymbol{g}}$ of the inverse generator directly (Eq. (2.3)).

**Compositional generalization.** We now formalize *compositional generalization*. Informally, compositional generalization is the ability to perceive out-of-domain images containing unseen concept combinations (e.g. "penguin" and "desert" in Fig. 1). To formalize this, we assume observed images $\mathcal{X}_{\text{ID}} \subset \mathcal{X}$ arise from only a subset of possible concept combinations $\mathcal{Z}_{\text{ID}} \subset \mathcal{Z}$, i.e., $\mathcal{X}_{\text{ID}} := \boldsymbol{f}(\mathcal{Z}_{\text{ID}})$ (see Fig. 2). OOD concept combinations $\mathcal{Z}_{\text{OOD}}$ are defined as the set of all unseen combinations of slots

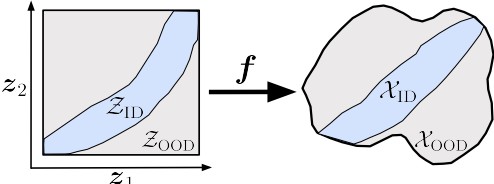

Figure 2: Visualization of a data generating process with in- and out-of-domain regions.

$$\mathcal{Z}_{\text{OOD}} := \{ \mathcal{Z}_1 \times \mathcal{Z}_2 \times \cdots \times \mathcal{Z}_K \} \setminus \mathcal{Z}_{\text{ID}} \quad \text{with} \quad \mathcal{Z}_k := \{ \boldsymbol{z}_k \in \mathbb{R}^m \mid \boldsymbol{z} \in \mathcal{Z}_{\text{ID}} \}, \tag{2.4}$$

which give rise to OOD images $\mathcal{X}_{\text{OOD}} := \boldsymbol{f}(\mathcal{Z}_{\text{OOD}})$ (Fig. 2). Compositional generalization is then achieved if Eq. (2.1) is satisfied both in-domain, for $\boldsymbol{z} \in \mathcal{Z}_{\text{ID}}$, and out-of-domain, for all $\boldsymbol{z} \in \mathcal{Z}_{\text{OOD}}$.

**The problem of identifiability.** Compositional generalization, using a generative or non-generative approach, requires identifying the ground-truth generator $\boldsymbol{f}$ or its inverse $\boldsymbol{g}$, both in-domain and out-of-domain. In-domain identifiability is a well-studied problem (Hyvärinen et al., 2023). It can be solved using both approaches by leveraging observed images $\boldsymbol{x} \in \mathcal{X}_{\text{ID}}$ together with self- (Gresele et al., 2021; von Kügelgen et al., 2021; Zimmermann et al., 2021) or weakly-supervised information (Hyvärinen and Morioka, 2016; Khemakhem et al., 2020; Locatello et al., 2020a) about the data-generating process. Out-of-domain identifiability, however, presents a different challenge: because $\boldsymbol{x} \in \mathcal{X}_{\text{OOD}}$ is unobserved, the strategies above cannot be applied. Consequently, out-of-domain identifiability must be implied by in-domain identifiability (Wiedemer et al., 2024b). This is only possible if $\boldsymbol{f}$ belongs to a function class $\mathcal{F}$ such that for all $\boldsymbol{f}^1, \boldsymbol{f}^2 \in \mathcal{F}$

$$\forall \boldsymbol{z} \in \mathcal{Z}_{\text{ID}}, \ \boldsymbol{f}^1(\boldsymbol{h}_\pi(\boldsymbol{z})) = \boldsymbol{f}^2(\boldsymbol{z}) \implies \forall \boldsymbol{z} \in \mathcal{Z}_{\text{OOD}}, \ \boldsymbol{f}^1(\boldsymbol{h}_\pi(\boldsymbol{z})) = \boldsymbol{f}^2(\boldsymbol{z}), \tag{2.5}$$

which equivalently implies that for all inverses $\boldsymbol{g}^1, \boldsymbol{g}^2 \in \mathcal{G} := \{ \boldsymbol{f}^{-1} \mid \boldsymbol{f} \in \mathcal{F} \}$

$$\forall \boldsymbol{x} \in \mathcal{X}_{\text{ID}}, \ \boldsymbol{h}_\pi(\boldsymbol{g}^1(\boldsymbol{x})) = \boldsymbol{g}^2(\boldsymbol{x}) \implies \forall \boldsymbol{x} \in \mathcal{X}_{\text{OOD}}, \ \boldsymbol{h}_\pi(\boldsymbol{g}^1(\boldsymbol{x})) = \boldsymbol{g}^2(\boldsymbol{x}). \tag{2.6}$$

If these implications do not hold then the problem is *non-identifiable*, since there is no way to distinguish between $\boldsymbol{f}^1$ and $\boldsymbol{f}^2$ or $\boldsymbol{g}^1$ and $\boldsymbol{g}^2$ from observed data.

**Further assumptions on $\boldsymbol{f}$.** Under our current assumptions, the ground-truth generator can be any diffeomorphism from $\mathcal{Z}$ to $\mathcal{X}$. This function class is far too large to satisfy Eq. (2.5). Thus, further assumptions on $\boldsymbol{f}$ are required. Recently, Brady et al. (2025, Thm. 4.4) proved that diffeomorphisms (on their image) with the following form will satisfy Eq. (2.5) (when $\boldsymbol{f}$ is sufficiently nonlinear)

$$\boldsymbol{f}(\boldsymbol{z}) = \sum_{k=1}^{K} \boldsymbol{f}^k\left(\boldsymbol{z}_k\right) + \sum_{\boldsymbol{\alpha}:|\boldsymbol{\alpha}|\leq n} \boldsymbol{c}_{\boldsymbol{\alpha}} \boldsymbol{z}^{\boldsymbol{\alpha}}, \tag{2.7}$$

where $n \in \mathbb{N}$, $\boldsymbol{c}_{\boldsymbol{\alpha}} \in \mathbb{R}^{d_x}$, and $\boldsymbol{\alpha} \in \mathbb{N}_0^{d_z}$ is a *multi-index*.[1] This function class, denoted $\mathcal{F}_{\text{int}}$, was introduced to model concepts with varying degrees of interaction $n$. For example, when $n = 1$, the second-sum on the RHS vanishes and concepts can only interact additively (Lachapelle et al., 2023). For $n > 1$ concepts can interact explicitly via polynomial functions of components from different slots. This aims to capture more complex concept interactions such as between objects and backgrounds. Such functions thus offer a flexible model of visual concepts, and are the largest function class shown to enable OOD identifiability (Eq. (2.5)). For these reasons, we assume that ground-truth generators $\boldsymbol{f}$ belong to $\mathcal{F}_{\text{int}}$, and inverse generators $\boldsymbol{g}$ to $\mathcal{G}_{\text{int}} := \{\boldsymbol{f}^{-1} \mid \boldsymbol{f} \in \mathcal{F}_{\text{int}}\}$.

**Guaranteeing compositional generalization.** We can now formalize what is required to guarantee compositional generalization using both a generative and non-generative approach. To this end, we assume ID identifiability holds for a decoder $\hat{\boldsymbol{f}}$ and encoder $\hat{\boldsymbol{g}}$, i.e., Equations 2.2 and 2.3 are satisfied in-domain. Since ground-truth generators $\mathcal{F}_{\text{int}}$ and inverses $\mathcal{G}_{\text{int}}$ satisfy Equations 2.5 and 2.6, OOD identifiability is guaranteed if the decoder class $\hat{\boldsymbol{f}} \in \mathcal{F}_{\text{dec}}$ is constrained to $\mathcal{F}_{\text{dec}} = \mathcal{F}_{\text{int}}$ and similarly if the encoder class $\hat{\boldsymbol{g}} \in \mathcal{G}_{\text{enc}}$ is constrained to $\mathcal{G}_{\text{enc}} = \mathcal{G}_{\text{int}}$. Compositional generalization is thus possible *in theory* for both generative and non-generative approaches. This does not necessarily imply, however, that it can be guaranteed *in practice* for both approaches. Specifically, guaranteeing compositional generalization in practice depends on whether practical means exist to enforce $\hat{\boldsymbol{f}} \in \mathcal{F}_{\text{int}}$ in the generative case and $\hat{\boldsymbol{g}} \in \mathcal{G}_{\text{int}}$ in the non-generative case.

# 3 THEORETICAL ANALYSIS

In this section, we theoretically analyze the structure of $\mathcal{F}_{\text{int}}$ and $\mathcal{G}_{\text{int}}$ to understand whether a model can be constrained to these classes with practical means such as regularization or architecture design.

**Structure of $\mathcal{F}_{\text{int}}$.** Generators in $\mathcal{F}_{\text{int}}$ are defined as diffeomorphisms which take the form of Eq. (2.7). Consequently, to enforce $\hat{\boldsymbol{f}} \in \mathcal{F}_{\text{int}}$, we must constrain a decoder to match this form. This can be done in a straightforward manner via architecture design. For example, the first term on the RHS of Eq. (2.7) can be parameterized as the sum of slot-wise neural networks and the second term using learned coefficients for $\boldsymbol{c}_{\boldsymbol{\alpha}}$. Furthermore, we highlight that functions of the form in Eq. (2.7) can equivalently be expressed as having block-diagonal derivative matrices $D^{n+1}\boldsymbol{f}(\boldsymbol{z})$ (Brady et al., 2025; Lachapelle et al., 2023). Specifically, if $n = 1$, then the Hessian $D^2 \boldsymbol{f}$ has the structure that for any two slots $\boldsymbol{z}_k$ and $\boldsymbol{z}_l$,

$$\forall 1 \leq k \neq l \leq K, \quad D_{\boldsymbol{z}_k} D_{\boldsymbol{z}_l} \boldsymbol{f}(\boldsymbol{z}) = 0. \tag{3.1}$$

For $n > 1$, analogous conditions hold for higher-order derivatives (Brady et al., 2025). Thus, we can also enforce that $\hat{\boldsymbol{f}} \in \mathcal{F}_{\text{int}}$ for a decoder $\hat{\boldsymbol{f}}$ via regularization. For example, when $n = 1$, we can use the following regularizer (with similar expressions for higher-order derivatives when $n > 1$)

$$\mathcal{R}(\hat{\boldsymbol{f}}, \boldsymbol{z}) = \sum_{k \neq l \in [K]} \left\| D_{\boldsymbol{z}_k, \boldsymbol{z}_l}^2 \hat{\boldsymbol{f}}(\boldsymbol{z}) \right\|. \tag{3.2}$$

## 3.1 STRUCTURE OF $\mathcal{G}_{\text{int}}$.

We now investigate the structure of inverse generators in $\mathcal{G}_{\text{int}}$. For simplicity, we present results for $n = 1$; similar statements can in principle be derived for higher order derivatives for the case $n > 1$.

---

[1] A *multi-index* is an ordered tuple $\boldsymbol{\alpha} = (\alpha_1, \alpha_2, ..., \alpha_d)$ of non-negative integers $\alpha_i \in \mathbb{N}_0$, with operations $|\boldsymbol{\alpha}| := \alpha_1 + \alpha_2 + ... + \alpha_d$, and $\boldsymbol{z}^{\boldsymbol{\alpha}} := z_1^{\alpha_1} z_2^{\alpha_2} ... z_d^{\alpha_d}$.

We first note that inverse generators $\boldsymbol{g} \in \mathcal{G}_{\text{int}}$ do not admit an analytical form similar to Eq. (2.7). Thus, understanding their structure requires analyzing finer-grained properties of these functions. To this end, we investigate their derivatives. We also study whether we can find architectures with an inductive bias towards $\mathcal{G}_{\text{int}}$, but delegate this to Appendix A.2 due to space constraints.

We will first assume that the observed dimension $d_x$ equals the ground-truth latent dimension $d_z$ such that $\mathcal{X} = \mathcal{Z}$. In this case, we show that, similar to generators in $\mathcal{F}_{\text{int}}$, inverse generators in $\mathcal{G}_{\text{int}}$ have a structured Jacobian and Hessian. Specifically, we prove the following result.

**Lemma 3.1.** *Let $\boldsymbol{g} \in \mathcal{G}_{\text{int}}$ for $n = m = 1$ and $d_x = d_z$. Then $\boldsymbol{g}$ has the property that for $\boldsymbol{x} \in \mathcal{X}$*

$$(D\boldsymbol{g})^{-\top}(\boldsymbol{x}) D^2 \boldsymbol{g}_s(\boldsymbol{x}) (D\boldsymbol{g})^{-1}(\boldsymbol{x}) \in \text{Diag}(d_x) \tag{3.3}$$

*is a diagonal matrix for $s \in [d_z]$. Further, if $\boldsymbol{g}$ is a diffeomorphism satisfying Eq. (3.3) then $\boldsymbol{g} \in \mathcal{G}_{\text{int}}$.*

Thus, when $\mathcal{X} = \mathcal{Z}$, enforcing that $\hat{\boldsymbol{g}} \in \mathcal{G}_{\text{int}}$ requires constraining an encoder according to Eq. (3.3). This is achievable, for instance, through regularization on the derivatives of $\hat{\boldsymbol{g}}$ analogous to Eq. (3.2).

This setting, however, is not applicable to image data since images typically lie in a manifold embedded in a higher-dimensional ambient space $\mathbb{R}^{d_x}$. We therefore consider the more practical case where $d_x$ is larger than the ground-truth latent dimension $d_z$. Specifically, we assume $d_x \geq d_z^3$. In this case, we first prove that the aforementioned structure on $D\boldsymbol{g}$ and $D^2\boldsymbol{g}$ is no longer present.

**Theorem 3.2.** *Assume that $d_x \geq d_z^3$. Let $\boldsymbol{B}_l \in \mathbb{R}^{d_x \times d_x}$ be symmetric matrices for $1 \leq l \leq d_z$. Then there is for any $\boldsymbol{x}_0 \in \mathbb{R}^{d_x}$ and for almost every $\boldsymbol{A} \in \mathbb{R}^{d_z \times d_x}$ a generator $\boldsymbol{f} \in \mathcal{F}_{\text{int}}$ with a (left)-inverse $\boldsymbol{g} \in \mathcal{G}_{\text{int}}$, such that $\boldsymbol{f}(0) = \boldsymbol{x}_0$ and $D\boldsymbol{g}(\boldsymbol{x}_0) = \boldsymbol{A}$ and $D^2\boldsymbol{g}_l(\boldsymbol{x}_0) = \boldsymbol{B}_l$ for $1 \leq l \leq d_z$.*

Thus, when $d_x \gg d_z$, $D^2\boldsymbol{g}$ and $D\boldsymbol{g}$ can be arbitrary matrices (up to a set of measure 0). We emphasize that this result applies to $\mathcal{F}_{\text{int}}$ with arbitrary interaction degree $n \geq 1$ and any slot dimensions. However, the structure expressed in Eq. (3.3) does not vanish entirely from $\boldsymbol{g}$. Instead, it persists, but only for the restriction of $\boldsymbol{g}$ to the data manifold $\mathcal{X}$. Specifically, the constraint Eq. (3.3) holds more generally for $n = m = 1$ when $D\boldsymbol{g}$ is projected on the tangent space $T_{\boldsymbol{x}}\mathcal{X}$ of the data manifold, i.e.,

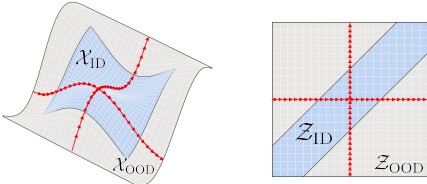

Figure 3: Structure of a data manifold $\mathcal{X}$ and latent manifold $\mathcal{Z}$.

$$\left( (D\boldsymbol{g}(\boldsymbol{x}) \Pi_{T_{\boldsymbol{x}}\mathcal{X}})^+ \right)^\top (\boldsymbol{z}) D^2 \boldsymbol{g}_s(\boldsymbol{x}) (D\boldsymbol{g}(\boldsymbol{x}) \Pi_{T_{\boldsymbol{x}}\mathcal{X}})^+ \in \text{Diag}(d_z) \tag{3.4}$$

where $\Pi_{T_{\boldsymbol{x}}\mathcal{X}}$ denotes the orthogonal projection on the tangent space (see Lemma A.4 for details).

Constraining an encoder such that $\hat{\boldsymbol{g}} \in \mathcal{G}_{\text{int}}$ thus requires enforcing this structure on $\hat{\boldsymbol{g}}$. This is challenging because the constraints depend on the geometry of the data manifold $\mathcal{X}$. Enforcing such constraints is thus not only impractical but also ill-posed since the geometry of out-of-domain regions $\mathcal{X}_{\text{OOD}} \subset \mathcal{X}$ is unobserved. This suggests that constraining an encoder through approaches such as architectural design or regularization is infeasible, as any such method would necessarily be data-dependent as well as implicitly assume knowledge of $\mathcal{X}_{\text{OOD}}$.

We contrast this with the reverse direction for $\boldsymbol{f} \in \mathcal{F}_{\text{int}}$. In this case, the structure to be enforced (see Eq. (3.1)) is not manifold-dependent but is always aligned with the global coordinate axes (Fig. 3, right). This allows for a universal procedure to constrain a decoder to $\mathcal{F}_{\text{int}}$, rather than a manifold-dependent one (Fig. 3, left). Moreover, such constraints can also be applied in OOD regions, since the manifold $\mathcal{Z}_{\text{ID}}$ extends in a Cartesian fashion and its structure is therefore known.

**Special case of $n = 0$.** We briefly discuss the case of functions in $\mathcal{F}_{\text{int}}$ when $n = 0$. These functions, introduced by Brady et al. (2023), are a special case of $n = 1$ with the additional, more restrictive condition $|D_{\boldsymbol{z}_k} \boldsymbol{f}_i(\boldsymbol{z})| \cdot |D_{\boldsymbol{z}_l} \boldsymbol{f}_i(\boldsymbol{z})| = 0$ for each $i \in [d_x]$. In other words, each pixel $i$ depends only on a single slot and no interactions (such as occlusions) between objects are possible. In this case, we can find a left inverse $\boldsymbol{g}$ of $\boldsymbol{f} \in \mathcal{F}_{\text{int}}^{n=0}$ (for any $d_x \geq d_z$) whose Jacobian satisfies the sparsity constraint $|D_{\boldsymbol{x}_i} \boldsymbol{g}_k| \cdot |D_{\boldsymbol{x}_j} \boldsymbol{g}_l| = 0$ for $l \neq k$. This additional structure can thus be leveraged to restrict $\mathcal{G}_{\text{enc}}$. However, this remains challenging in practice because the sparsity pattern (i.e., which slots $\boldsymbol{z}_l$ depends on which pixel $\boldsymbol{x}_i$) is not known a-priori. In Section 5, we study whether concepts satisfying $n = 0$ can empirically enable compositional generalization for non-generative approaches.

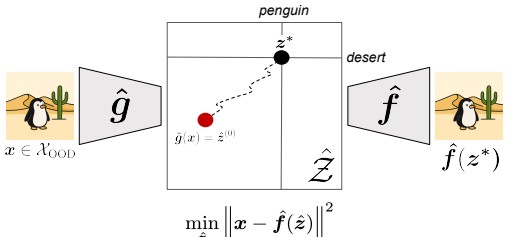 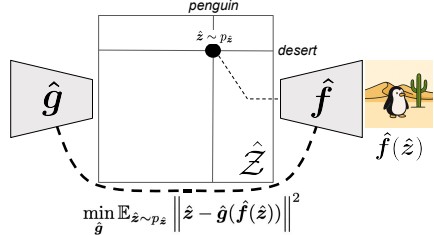

Figure 4: **Approaches for inverting a generator out-of-domain.** *Left.* Visualization of gradient-based search to invert a decoder $\hat{f}$ out-of-domain, with initialization given by an encoder $\hat{g}$. *Right.* Visualization of generative replay in which an encoder is trained on OOD images generated by a decoder.

**Takeaways.** Our results suggest that it is generally not feasible to constrain an encoder class such that $\mathcal{G}_{enc} = \mathcal{G}_{int}$. This means that there can exist encoders $\hat{g} \in \mathcal{G}_{enc}$ that satisfy Eq. (2.3) in-domain, but fail to generalize this behavior OOD (see Fig. 1). Consequently, for non-generative methods, whether compositional generalization occurs depends on whether the optimization process happens to avoid converging to such a solution. In contrast, generative methods can avoid such solutions by construction through appropriate inductive biases on a decoder. In Sec. 5, we investigate the practical consequences of this asymmetry on compositional generalization for both classes of methods.

## 4 SEARCH AND REPLAY

Our results in Sec. 3 suggest that guaranteeing compositional generalization requires a generative approach, i.e., inverting a learned decoder $\hat{f}$. If a decoder admits an explicit inverse, this inversion is trivial. For image data, however, constructing such a decoder is challenging as this generally requires that $\mathcal{X} = \mathbb{R}^{d_x}$ (Papamakarios et al., 2021). Consequently, inverting $\hat{f}$ requires solving an inference problem: given an image $x$, we must find a latent $z^*$ such that

$$x = \hat{f}(z^*). \tag{4.1}$$

In this section, we explore strategies for solving this inference problem efficiently.

**Inversion on $\mathcal{X}_{ID}$.** For in-domain images, i.e. $x \in \mathcal{X}_{ID}$, inverting a decoder $\hat{f}$ to obtain $z^*$ can be done directly by training an *autoencoder*. Specifically, we can leverage an encoder $\hat{g}$ to invert $\hat{f}$ in-domain by minimizing the reconstruction objective

$$\min_{\hat{f}, \hat{g}} \mathbb{E}_{x \sim \mathcal{X}_{ID}} \left\| x - \hat{f}(\hat{g}(x)) \right\|^2. \tag{4.2}$$

Thus, for images $x \in \mathcal{X}_{ID}$, $z^*$ (Eq. (4.1)) can be obtained directly as the output of the encoder. For out-of-domain images, however, minimizing Eq. (4.2) is not an option since $x \in \mathcal{X}_{OOD}$ is unobserved. Thus, to efficiently solve Eq. (4.1) on $\mathcal{X}_{OOD}$, other strategies are required. We explore two such strategies: gradient-based search (Sec. 4.1) and generative replay (Sec. 4.2).

### 4.1 GRADIENT-BASED SEARCH.

We note that the inference problem in Eq. (4.1) can be expressed as an optimization problem, i.e.,

$$z^* = \arg\min_{\hat{z}} \left\| x - \hat{f}(\hat{z}) \right\|^2. \tag{4.3}$$

Thus, for OOD images $x \in \mathcal{X}_{OOD}$, we can recover $z^*$ *online* by solving Eq. (4.3) using *gradient-based optimization*. The efficiency of this, however, depends on the initialization $\hat{z}^{(0)}$. If $\hat{z}^{(0)}$ is far from the optimum, many gradient steps are required, leading to slow or suboptimal convergence. To mitigate this, we can leverage the encoder trained on $\mathcal{X}_{ID}$ to provide an initial prediction for $z^*$ such that $\hat{z}^{(0)} = \hat{g}(x)$ and then optimize Eq. (4.3) (see Fig. 4, left). Intuitively, the encoder gives a fast "System 1" guess that constrains the space for slower, "System 2" reasoning (Kahneman, 2011; Prabhudesai et al., 2023a), where "reasoning" corresponds to gradient-based search (LeCun, 2022) (for pseudocode, see Fig. 8, top).

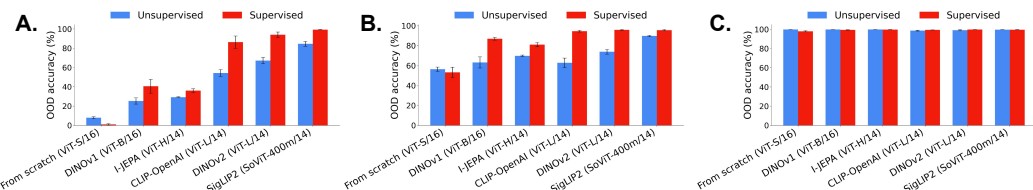

Figure 5: **OOD performance for non-generative methods.** We report OOD performance across three dataset splits for non-generative methods trained with and without supervision and with differing base encoders. On PUG-Background (**A.**), we see that strong OOD performance generally emerges only for base encoders with large scale pretraining such as SigLIP2 and is otherwise poor. We see a similar trend on PUG-Texture (**B.**) though OOD performance is generally higher across models. On PUG-Object (**C.**), concepts do not interact, such that $\mathcal{G}_{\text{int}}$ is more constrained (Sec. 3.1). This structure is sufficient for all models to generalize OOD.

## 4.2 GENERATIVE REPLAY

For out-of-domain images, Eq. (4.1) can also be solved in an *offline* manner by leveraging *generative replay* (Kurth-Nelson et al., 2023; Schwartenbeck et al., 2023). Recall that images $x \in \mathcal{X}_{\text{OOD}}$ are generated by $f$ as combinations of ground-truth slots $z_k$. Since the decoder $\hat{f}$ identifies $f$ up to slot-wise transformations, images $x \in \mathcal{X}_{\text{OOD}}$ can likewise be generated by re-combining inferred slots $\hat{z}_k$. Concretely, this can be achieved by sampling a latent $\hat{z}$ from a distribution $p_{\hat{z}}$ with independent slot-wise marginals and decoding them with $\hat{f}$ such that $\hat{f}(\hat{z}) \in \mathcal{X}_{\text{OOD}}$. We can then solve Eq. (4.1) out-of-domain by training an encoder $\hat{g}$ on these samples such that $\hat{g}(\hat{f}(\hat{z})) = \hat{z}$ (see Fig. 4, right). This is captured by the following objective function (Wiedemer et al., 2024a) (see Fig. 8, bottom)

$$\min_{\hat{g}} \mathbb{E}_{\hat{z} \sim p_{\hat{z}}} \left\| \hat{z} - \hat{g}(\hat{f}(\hat{z})) \right\|^2 . \tag{4.4}$$

## 5 EXPERIMENTS

In this section, we conduct an experimental study with two main components. First, we aim to assess the extent to which non-generative methods can achieve compositional generalization in practice without enforcing explicit constraints to this end. Second, we evaluate whether generative methods, which leverage search (Sec. 4.1) and replay (Sec. 4.2), can achieve superior compositional generalization. We describe our experimental setup below, further details can be found in App. B.

### 5.1 SETUP

**Data.** We are interested in evaluating compositional generalization for images in realistic settings. This is challenging, however, since web-scale image datasets do not provide explicit controllability over in- and out-of-domain regions. To address this, we leverage the PUG datasets (Bordes et al., 2023), which offer photorealistic images while remaining explicitly controllable. The images we consider are defined by a background and one or two animals, which can take on 10 and 32 different values, respectively. In addition, animals can vary in position and texture.

Using this dataset, we construct three different in- and out-of-domain splits (see Fig. 7, left). In *PUG-Background*, $\mathcal{X}_{\text{OOD}}$ contains unseen combinations of animals and backgrounds. In *PUG-Texture*, $\mathcal{X}_{\text{OOD}}$ contains unseen combinations of animals and textures. Finally, in *PUG-Object*, $\mathcal{X}_{\text{OOD}}$ contains unseen combinations of animals. In this case, animals never occlude each other and therefore do not interact, meaning that concepts satisfy $n = 0$.

**Evaluating compositional generalization.** To evaluate compositional generalization on this data, we assume a model gives inferred latent slots $\hat{z}_k$. Each slot should encode either one of the two animals or the background, both on $\mathcal{X}_{\text{ID}}$ and $\mathcal{X}_{\text{OOD}}$. To test this, we train a slot-wise readout in-domain to predict the category of the corresponding animal or background. We then report out-of-domain accuracy for these predictions.

**Encoders.** We consider encoder architectures with the following structure. Images are first divided into patches and processed by a *base encoder*, which produces a set of embeddings. These embeddings are then mapped to slots by a *slot encoder* (see Fig. 7, right). We implement the base

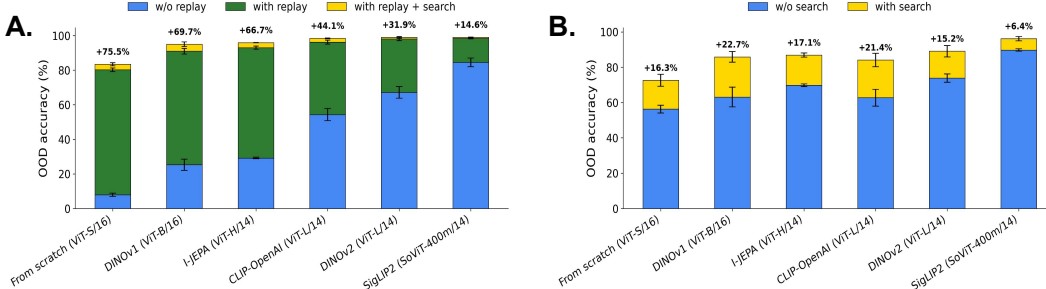

Figure 6: **OOD performance for generative methods.** We report OOD performance on PUG-Background and PUG-Texture for unsupervised autoencoders which leverage replay (Sec. 4.2) and search (Sec. 4.1) trained with differing base encoders. On PUG-Background (**A.**), we observe a significant increase in OOD performance using replay and additional gains through search. On PUG-Texture (**B.**) we also see a noticeable increase in OOD performance when using search.

encoder using a Vision Transformer (ViT) (Dosovitskiy et al., 2020), while the slot encoder is either a Transformer (Vaswani et al., 2017) or a Slot Attention module (Locatello et al., 2020b).

Ideally, we would train encoders from scratch using state-of-the-art non-generative methods. However, such methods rely on large-scale datasets, while our datasets are comparatively small ($\sim$20000 images). We thus leverage pretrained models. Concretely, for the base encoder, we use DINOv1 (Caron et al., 2021), I-JEPA (Assran et al., 2023), DINOv2 (Oquab et al., 2024), CLIP (Radford et al., 2021), and SigLIP2 (Tschannen et al., 2025). These models are optionally fine-tuned using a LoRA adapter (Hu et al., 2022), while the slot encoder is always trained from scratch. We note that the PUG datasets were not contained in the pretraining set for these models, thus data contamination is not an issue. Finally, we also include a ViT-Small base encoder trained from scratch.

**Decoders.** Brady et al. (2025) argued that constraining a decoder to $\mathcal{F}_{\text{int}}$ can be done approximately using a regularized cross-attention Transformer. In this model, pixels query slots, and a regularization term is applied to the resulting attention weights to encourage pixels to specialize to a single slot. This model is also sufficiently flexible to capture complex images and concepts with varying degrees of interaction. For these reasons, we leverage such decoders in our experiments. In § C, we also report results when using unstructured decoders which are not designed to match $\mathcal{F}_{\text{int}}$.

**Training objectives.** To learn a representation $\hat{z}$, we train non-generative methods using both a supervised and unsupervised objective. In the supervised setting, the encoder is trained on $\mathcal{X}_{\text{ID}}$ to predict the animal and background categories using a cross-entropy loss. In the unsupervised case, we train a variational autoencoder (VAE) (Kingma and Welling, 2014) with our regularized decoder architecture. This case is nevertheless non-generative since the encoder is only constructed to invert the decoder on $\mathcal{X}_{\text{ID}}$, and not on $\mathcal{X}_{\text{OOD}}$. For our generative methods, we take this learned decoder and invert it on $\mathcal{X}_{\text{OOD}}$ using search and replay.

### 5.2 RESULTS

**Non-generative methods.** In Fig 5, we evaluate compositional generalization for non-generative methods trained on each PUG split. All methods achieve nearly perfect ID accuracy ($\sim 99\%$), thus we only visualize OOD accuracy. For each base encoder, we report the OOD accuracy obtained with the best-performing combination of slot encoder and fine-tuning choice. In Fig. 5 A. (PUG-Background), base encoders trained from scratch (ViT-Small) or pretrained on relatively small corpora (e.g., DINOv1 on ImageNet) fail to generalize OOD. OOD accuracy improves with encoders leveraging larger-scale pretraining, such as SigLIP2. In Fig. 5 B. (PUG-Texture), we observe a boost in OOD performance across models, though performance remains suboptimal overall. Again, models with larger-scale pretraining exhibit stronger OOD performance.

Finally, in Fig. 5 C., we report results for PUG-Object in which concepts do not interact. This corresponds to the special case of $n = 0$ in Sec. 3 in which $\mathcal{G}_{\text{int}}$ is more structured. Although we do not explicitly enforce this structure on the models, they nevertheless achieve near-perfect OOD accuracy, indicating that such structure makes compositional generalization fundamentally easier.

**Generative Methods.** In Fig. 6, we take the autoencoders trained in Fig. 5 and report OOD accuracy after leveraging replay and search for inverting the decoder. On PUG-Background (Fig. 6 A.), we observe a significant increase in OOD accuracy when training encoders with replay across all models, with further improvement when additionally using search. On PUG-Texture (Fig. 6 B.), replay cannot be applied, since in our setup, slots are designed to capture objects and backgrounds, and therefore cannot be trivially recomposed to generate novel animal–texture combinations. However, leveraging search yields a clear improvement in OOD performance across all models. We do not report results on PUG-Object as all non-generative methods achieve near-perfect OOD performance on this dataset. Thus, further OOD gains through search and replay are not possible.

## 6 RELATED WORK

**Limitations of non-generative methods for compositional generalization.** Several empirical studies have shown limitations in compositional generalization for non-generative methods trained using natural language supervision (Assouel et al., 2025; Lewis et al., 2022; Ma et al., 2023; Tong et al., 2024; Yuksekgonul et al., 2022). These works generally posit that poor generalization arises from issues with standard contrastive language-image training objectives. In contrast, our theoretical and empirical contributions suggest that such issues are more fundamental, arising from the structure of the inverses of the unknown generator, i.e.,. $\mathcal{G}_{\text{int}}$.

**Generative approaches for improving generalization.** The idea that a generative approach can enable compositional generalization has long been advocated in the cognitive science community (Lake et al., 2015; 2017; Tenenbaum et al., 2011). Empirical realizations of this idea have recently been shown for diffusion models repurposed as classifiers (Jeong et al., 2025; Wang et al., 2025). Further (Prabhudesai et al., 2023a;b), showed that inverting a generative model with mechanisms similar to gradient-based search (Sec. 4.1) improves object-decomposition for OOD images and enhances the robustness of classifiers. Recent work explored training encoder-only models using synthetically generated data similar to Sec. 4.2, showing improvements in representations (Fan et al., 2025; Tian et al., 2023) and compositional generalization (Assouel et al., 2022; Jung et al., 2024; Wiedemer et al., 2024a). Our work provides a theoretical motivation for these approaches by highlighting challenges in achieving compositional generalization using non-generative methods.

**Causal and anti-causal learning.** Our theoretical contribution relates to ideas in the field of causality. A key heuristic in this area posits that the factorization $P(\text{cause})P(\text{effect}|\text{cause})$ is, in general, less complex than the reverse factorization $P(\text{effect})P(\text{cause}|\text{effect})$ (Janzing and Schölkopf, 2010; Sun et al., 2006; 2008). It was conjectured by Kilbertus et al. (2018) that this principle indicates generalization is typically easier to achieve in the causal direction than in the anti-causal direction. Moreover, they propose an abstract version of the search procedure (Sec. 4.1). The present paper can be seen as providing a formal justification for these ideas through theoretical insights on the structure of generators $\boldsymbol{f}$ (the causal direction) and their inverses $\boldsymbol{g}$ (the anti-causal direction).

## 7 DISCUSSION

**Limitations.** Our theory is limited to generators which belong to $\mathcal{F}_{\text{int}}$. We studied this function class as it provides a suitable model of visual data and is the largest class which enables OOD identifiability. However, these results may, in principle, fail to generalize to function classes associated with other settings, where non-generative strategies may be effective. Additionally, while our experiments leverage photorealistic data, they focus on concepts in simple settings which do not fully capture the complexity of real world data. To this end, an important future question is to understand how to create benchmarks to evaluate compositional generalization in a rigorous manner on data at a more realistic scale. For further discussions on limitations, see § D.

**Conclusion.** In this work, we sought a principled understanding of whether compositional generalization should be pursued through generative or non-generative approaches. Theoretically, we showed that for non-generative methods, enforcing the structure needed to guarantee compositionality tends to be infeasible. As a result, generalization is determined largely by the optimization process rather than by principled guarantees. Empirically, we observed that methods optimized from scratch or with little pretraining data tend to fail at compositional generalization, while larger-scale pretrained models improve OOD performance at the cost of data efficiency. By contrast, generative

approaches can directly enforce constraints for compositional generalization which manifest in significant gains in OOD performance in practice. While scaling such generative approaches to more challenging settings remains an open problem, we hope our findings will inspire renewed interest in this direction.

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

# Appendices

## Table of Contents

**Use of Language Models**   Large language models (LLMs) were employed exclusively during the final stages of manuscript preparation for the purpose of refining language, grammar, and readability. They were not used for generating ideas, conducting analysis, or contributing to the substantive content of this work.

## A   PROOFS

In this section we collect the proofs of the results in the paper and some additional background material. First, in Section A.1 we investigate the local restrictions that $g \in \mathcal{G}_{\text{int}}$ need to satisfy. Similarly we investigate in Section A.2 whether we can enforce $g \in \mathcal{G}_{\text{int}}$ by architectural constraints. Let us, however, first introduce a notation for a subset of $\mathcal{F}_{\text{int}}$.

**Definition A.1** (Additive functions). We denote the function class of coordinate-wise additive functions $\boldsymbol{f} : \mathbb{R}^{d_z} \to \mathbb{R}^{d_x}$ by $\mathcal{F}_{\text{add}}$. They can be expressed as

$$\boldsymbol{f}(\boldsymbol{x}) = \sum_{i=1}^{d_z} \boldsymbol{f}_i(\boldsymbol{x}_i) \tag{A.1}$$

where $\boldsymbol{f}_i : \mathbb{R} \to \mathbb{R}^{d_x}$.

Clearly $\mathcal{F}_{\text{add}}$ agrees with $\mathcal{F}_{\text{int}}$ for $n = m = 1$, i.e., interactions of first order and blocks of dimension 1 and generally $\mathcal{F}_{\text{add}} \subset \mathcal{F}_{\text{int}}$ for $n \geq 1$ (higher order interactions and larger blocks are more flexible).

### A.1   LOCAL REGULARIZATION OF ENCODERS

As discussed in the main text, we can enforce $\boldsymbol{f} \in \mathcal{F}_{\text{int}}$ by enforcing that certain derivatives of $\boldsymbol{f}$ vanish (see equation 3.2). We now study to what extend this generalizes to functions $g \in \mathcal{G}_{\text{int}}$ that are left inverses of such functions.

**The key relation.**   The key relation that we need for the proofs below is that if $\boldsymbol{g} \circ \boldsymbol{f}(\boldsymbol{z}) = \boldsymbol{z}$ for two functions $\boldsymbol{f} : \mathbb{R}^{d_z} \to \mathbb{R}^{d_x}$ and $\boldsymbol{g} : \mathbb{R}^{d_x} \to \mathbb{R}^{d_z}$, then for every $s \in [d_z]$

$$D\boldsymbol{f}^{\top}(\boldsymbol{z})D^2\boldsymbol{g}_s(\boldsymbol{f}(\boldsymbol{z}))D\boldsymbol{f}(\boldsymbol{z}) + \sum_{k=1}^{d_x}(\partial_k \boldsymbol{g}_s)(\boldsymbol{f}(\boldsymbol{z}))D^2\boldsymbol{f}_k(\boldsymbol{z}) = 0. \tag{A.2}$$

This relation follows by straightforward calculation, indeed we find using the chain rule

$$
\partial_i \partial_j \boldsymbol{g}_s(\boldsymbol{f}(\boldsymbol{z})) = \partial_i \left( \sum_{k=1}^{d_x} \partial_j \boldsymbol{f}_k(\boldsymbol{z})(\partial_k \boldsymbol{g}_s)(\boldsymbol{f}(\boldsymbol{z})) \right)
$$

$$
= \sum_{k,l=1}^{d_x} \partial_j \boldsymbol{f}_k(\boldsymbol{z}) \partial_i \boldsymbol{f}_l(\boldsymbol{z})(\partial_k \partial_l \boldsymbol{g}_s)(\boldsymbol{f}(\boldsymbol{z})) + \sum_{k=1}^{d_x} \partial_i \partial_j \boldsymbol{f}_k(\boldsymbol{z})(\partial_k \boldsymbol{g}_s)(\boldsymbol{f}(\boldsymbol{z})) \tag{A.3}
$$

which is equation A.2 after rewriting the relation in matrix form.

**Restrictions for $d_x = d_z$.** We now prove Lemma 3.1 showing that for $d_x = d_z$, i.e., for equal dimension of latent space and data it is possible to find a local constraint for the inverses of additive functions $\boldsymbol{f} \in \mathcal{F}_{\mathrm{add}}$.

**Lemma 3.1.** *Let $\boldsymbol{g} \in \mathcal{G}_{\mathrm{int}}$ for $n = m = 1$ and $d_x = d_z$. Then $\boldsymbol{g}$ has the property that for $\boldsymbol{x} \in \mathcal{X}$*

$$
(D\boldsymbol{g})^{-\top}(\boldsymbol{x}) D^2 \boldsymbol{g}_s(\boldsymbol{x})(D\boldsymbol{g})^{-1}(\boldsymbol{x}) \in \mathrm{Diag}(d_x) \tag{3.3}
$$

*is a diagonal matrix for $s \in [d_z]$. Further, if $\boldsymbol{g}$ is a diffeomorphism satisfying Eq. (3.3) then $\boldsymbol{g} \in \mathcal{G}_{\mathrm{int}}$.*

*Remark* A.2. For higher dimensional slots there is a natural generalization, namely, the expression $D\boldsymbol{g}^{-\top} D^2 \boldsymbol{g}_s D\boldsymbol{g}^{-1}$ has a block diagonal structure.

*Proof.* Note that $\boldsymbol{g} \circ \boldsymbol{f}(\boldsymbol{z}) = \boldsymbol{z}$ implies $\mathrm{Id}_{d_z} = (D\boldsymbol{g} \circ \boldsymbol{f}) D\boldsymbol{f}$ and thus $D\boldsymbol{f}(z) = (D\boldsymbol{g})^{-1}(\boldsymbol{f}(\boldsymbol{z}))$. Therefore, we find using equation A.2

$$
(D\boldsymbol{g})^{-\top}(\boldsymbol{f}(\boldsymbol{z})) D^2 \boldsymbol{g}_s(\boldsymbol{f}(\boldsymbol{z}))(D\boldsymbol{g})^{-1}(\boldsymbol{f}(\boldsymbol{z})) = -\sum_{k=1}^{d_x}(\partial_k \boldsymbol{g}_s)(\boldsymbol{f}(\boldsymbol{z})) D^2 \boldsymbol{f}_k(\boldsymbol{z}) \in \mathrm{Diag}(d_z). \tag{A.4}
$$

where we used that $\boldsymbol{f} \in \mathcal{F}_{\mathrm{add}}$ implies that the off-diagonal entries of $D^2 \boldsymbol{f}$ vanish. This implies the first part of the statement. For the reverse statement, we apply equation A.2 to $\boldsymbol{f} \circ \boldsymbol{g}(\boldsymbol{x}) = \boldsymbol{x}$ (here we use $d_z = d_x$) and we find that

$$
0 = (D\boldsymbol{g})^{\top}(\boldsymbol{x}) D^2 \boldsymbol{f}_s(\boldsymbol{g}(\boldsymbol{x})) D\boldsymbol{g}(\boldsymbol{x}) + \sum_{k=1}^{d_z}(\partial_k \boldsymbol{f}_s)(\boldsymbol{g}(\boldsymbol{x})) D^2 \boldsymbol{g}_k(\boldsymbol{x}). \tag{A.5}
$$

We multiply this relation from the left and right by $(D\boldsymbol{g})^{-\top}(\boldsymbol{x})$ and $(D\boldsymbol{g})^{-1}(\boldsymbol{x})$ respectively (the inverses exist by assumption) and we find

$$
D^2 \boldsymbol{f}_s(\boldsymbol{g}(\boldsymbol{x})) = -\sum_{k=1}^{d_z}(\partial_k \boldsymbol{f}_s)(\boldsymbol{g}(\boldsymbol{x}))(D\boldsymbol{g})^{-\top}(\boldsymbol{x}) D^2 \boldsymbol{g}_k(\boldsymbol{x})(D\boldsymbol{g})^{-1}(\boldsymbol{x}) \in \mathrm{Diag}(d_z). \tag{A.6}
$$

Here we used the assumption equation 3.3 to conclude that the right hand side is diagonal. Therefore $\boldsymbol{f}$ has a diagonal Hessian which implies that it is additive. $\square$

The previous statement can be generalized to the general case $d_x > d_z$. The crucial ingredient is the following simple and standard lemma.

**Lemma A.3.** *Let $\boldsymbol{A} \in \mathbb{R}^{d_1 \times d_2}$ and $\boldsymbol{B} \in \mathbb{R}^{d_2 \times d_1}$ two matrices with $d_2 \geq d_1$ and assume that $\boldsymbol{AB} = \mathbf{1}_{d_1 \times d_1}$. Then $\boldsymbol{B} = (\boldsymbol{A}\Pi)^+$ where $\Pi$ denotes the orthogonal projection onto $\mathrm{Range}(\boldsymbol{B})$ and $(\cdot)^+$ the Moore-Penrose inverse of a matrix.*

*Proof.* We check that $\boldsymbol{B}$ satisfies the Moore-Penrose axioms ($\boldsymbol{MM}^+\boldsymbol{M} = \boldsymbol{M}$, $\boldsymbol{M}^+\boldsymbol{MM}^+ = \boldsymbol{M}^+$, $\boldsymbol{M}^+\boldsymbol{M}$ and $\boldsymbol{MM}^+$ are Hermitian). We find

$$
\boldsymbol{A}\Pi\boldsymbol{BA}\Pi = \boldsymbol{ABA}\Pi = \boldsymbol{A}\Pi \tag{A.7}
$$

where we used $\Pi\boldsymbol{B} = \boldsymbol{B}$ by definition of $\Pi$. Similarly, we obtain

$$
\boldsymbol{BA}\Pi\boldsymbol{B} = \Pi\boldsymbol{B} = \boldsymbol{B}. \tag{A.8}
$$

Next we claim that

$$\boldsymbol{B}\boldsymbol{A}\Pi = \Pi \tag{A.9}$$

which is Hermitian. Consider $v \in \mathbb{R}^{d_2}$ then by definition of $\Pi$ there is $w \in \mathbb{R}^{d_1}$ such that $\boldsymbol{B}w = \Pi v$ and thus

$$\boldsymbol{B}\boldsymbol{A}\Pi v = \boldsymbol{B}\boldsymbol{A}\boldsymbol{B}w = \boldsymbol{B}w = \Pi v. \tag{A.10}$$

Finally, we find

$$\boldsymbol{A}\Pi\boldsymbol{B} = \boldsymbol{A}\boldsymbol{B} = \boldsymbol{1}_{d_1 \times d_1}. \tag{A.11}$$

$\square$

We have the following generalization of Lemma 3.1.

**Lemma A.4.** *Let $\boldsymbol{f} \in \mathcal{F}_{\mathrm{add}}$ and $\boldsymbol{g}$ a left-inverse of $\boldsymbol{f}$. Then $\boldsymbol{g}$ has the property that for $\boldsymbol{x} \in \mathcal{X}$*

$$\left( (D\boldsymbol{g}(\boldsymbol{x})\Pi_{T_{\boldsymbol{x}}\mathcal{X}})^+ \right)^\top (\boldsymbol{z}) D^2\boldsymbol{g}_s(\boldsymbol{x}) \, (D\boldsymbol{g}(\boldsymbol{x})\Pi_{T_{\boldsymbol{x}}\mathcal{X}})^+ \in \mathrm{Diag}(d_z) \tag{A.12}$$

*is a diagonal matrix for $s \in [d_z]$. Here, we denote by $\Pi_{T_{\boldsymbol{x}}\mathcal{X}}$ the orthogonal projection on the tangent space at $\boldsymbol{x}$.*

*Proof.* Starting from equation A.2 we find that for $\boldsymbol{f} \in \mathcal{F}_{\mathrm{add}}$ we get

$$D\boldsymbol{f}^\top(\boldsymbol{z}) D^2\boldsymbol{g}_s(\boldsymbol{f}(\boldsymbol{z})) D\boldsymbol{f}(\boldsymbol{z}) \in \mathrm{Diag}(d_z). \tag{A.13}$$

Applying Lemma A.3 we find

$$D\boldsymbol{f}(\boldsymbol{z}) = \left( D\boldsymbol{g}(\boldsymbol{f}(\boldsymbol{z}))\Pi_{T_{\boldsymbol{f}(\boldsymbol{z})}\mathcal{X}} \right)^+ \tag{A.14}$$

because the range of $D\boldsymbol{f}$ is the tangent space of the data manifold. Therefore we conclude that for $\boldsymbol{x} \in \mathcal{X}$ the relation equation A.12 indeed holds. $\square$

**Regularization for $d_x > d_z$.** In this paragraph we investigate the local restrictions that $\boldsymbol{g} \in \mathcal{G}_{\mathrm{int}}$ need to satisfy, and in particular we prove Theorem 3.2. The proof of Theorem 3.2 requires two lemmas as a key ingredient, which state that the crucial constraint on the second derivative stated in equation A.2 can be satisfied for a suitable choice of $\boldsymbol{M} = D\boldsymbol{f}(0)$ and $D^2\boldsymbol{f}(0)$ for given matrices $\boldsymbol{B}_s$ corresponding to the Hessian of $\boldsymbol{g}$ and almost every matrix $\boldsymbol{A}$ (corresponding to the Jacobian of $\boldsymbol{g}$). The first lemma establishes the existence of $\boldsymbol{M}$ such that first term in equation A.2 (given by $\boldsymbol{M}^\top \boldsymbol{B}_s \boldsymbol{M}$ is diagonal for all $s$. The second lemma constructs suitable second derivatives $D^2\boldsymbol{f}$ so that the relation equation A.2 also holds for the diagonal entries.

**Lemma A.5.** *Assume $d_x \geq d_z^3$. For all symmetric matrices $\boldsymbol{B}_s \in \mathbb{R}^{d_x \times d_x}$ for $s \in [d_z]$, and almost every $\boldsymbol{A} \in \mathbb{R}^{d_z \times d_x}$ there is a matrix $\boldsymbol{M} \in \mathbb{R}^{d_x \times d_z}$ such that $\boldsymbol{M}^\top \boldsymbol{B}_s \boldsymbol{M} \in \mathrm{Diag}(d_z)$ for $s \in [d_z]$ and $\boldsymbol{A}\boldsymbol{M} = \mathrm{Id}_{d_z}$.*

*Remark A.6.*      1. Counting parameters and equations, we find that $\boldsymbol{M}$ has $d_z d_x$ parameters and (by symmetry of $\boldsymbol{B}_s$) there are

$$d_z \cdot \frac{d_z(d_z - 1)}{2} + d_z^2 = \frac{d_z^2(d_z + 1)}{2} \tag{A.15}$$

equations. So, generally, we expect the result to hold for $d_x \geq d_z(d_z + 1)/2$.

2. On the other hand, the result does not hold for every $\boldsymbol{A}$ with maximal rank. Indeed, there can be a non-trivial null set of full rank matrices $\boldsymbol{A}$ such that the result does not hold. E.g., consider $d_z = 2$, $\boldsymbol{A} \in \mathbb{R}^{d_z \times d_x}$ such that all entries of $\boldsymbol{A}$ are zero except $\boldsymbol{A}_{1,1} = \boldsymbol{A}_{2,2} = 1$. Moreover, $\boldsymbol{B}_1$ has all entries zero except $(\boldsymbol{B}_1)_{1,2} = (\boldsymbol{B}_1)_{2,1} = 1$. Then $\boldsymbol{A}\boldsymbol{M} = \mathrm{Id}_{d_z}$ implies that $\boldsymbol{M}_{1,1} = \boldsymbol{M}_{2,2} = 1$, and $\boldsymbol{M}_{1,2} = \boldsymbol{M}_{2,1} = 0$. But then we find $\boldsymbol{M}_{:,1}^\top \boldsymbol{B}_1 \boldsymbol{M}_{:,2} = (\boldsymbol{B}_1)_{1,2} = 1 \neq 0$.

*Proof.* We inductively construct $d_z$ linear subspaces $V_i \subset \mathbb{R}^{d_x}$ such that $\dim(V_i) = d_z$ and

$$(\boldsymbol{v}^i)^\top \boldsymbol{B}_s \boldsymbol{v}^j = 0 \tag{A.16}$$

for $\boldsymbol{v}^i \in V_i$, $\boldsymbol{v}^j \in V_j$ and $i \neq j$. We pick $V_1$ arbitrarily. Then, given a basis $\boldsymbol{v}^{i,1}, \ldots, \boldsymbol{v}^{i,d_z}$ of $V_i$ for $i \leq j$ we select $V_{j+1} \subset \ker \boldsymbol{T}_j$ where $\boldsymbol{T}_j : \mathbb{R}^{d_x} \to \mathbb{R}^{d_z^2 \cdot j}$ given by $(\boldsymbol{T}_j \boldsymbol{v})_{s,(k,i)} = (\boldsymbol{v}^{i,k})^\top \boldsymbol{B}_s \boldsymbol{v}$ (here it is convenient to identify $[d_z^2 \cdot j]$ with $[d_z] \times ([d_z] \times [j])$). By assumption $d_x - d_z^2 \cdot j \geq d_x - d_z^2 \cdot (d_z - 1) \geq d_z$ and therefore $\dim \ker \boldsymbol{T}_j \geq d_z$ and we can find a suitable subspace $V_{j+1} \subset \ker \boldsymbol{T}_j$. Given a matrix $\boldsymbol{A} = (\boldsymbol{a}^1, \ldots, \boldsymbol{a}^{d_z})^\top \in \mathbb{R}^{d_z \times d_x}$, we want to find $\boldsymbol{w}^i \in V_i$ so that $\boldsymbol{M} = (\boldsymbol{w}^1, \ldots, \boldsymbol{w}^{d_z})$ satisfies $\boldsymbol{A} \boldsymbol{M} = \mathrm{Id}_{d_z}$. Equivalently $\boldsymbol{A} \boldsymbol{w}^i = \boldsymbol{e}^i$, where $\boldsymbol{e}^i$ denotes the $i$-th standard basis vector. We expand into the basis of $V_i$, i.e., $\boldsymbol{w}^i = \sum_j \boldsymbol{\lambda}_j^i \boldsymbol{v}^{i,j}$ and find the equivalent relation

$$\boldsymbol{A} \boldsymbol{w}^i = (\boldsymbol{a}^1, \ldots, \boldsymbol{a}^{d_z})^\top (\boldsymbol{v}^{i,1}, \ldots, \boldsymbol{v}^{i,d_z}) \boldsymbol{\lambda}^i = \boldsymbol{e}^i. \tag{A.17}$$

Since the second matrix has maximal rank $((\boldsymbol{v}^{i,k})_{1 \leq k \leq d_z}$ is a basis of $V_i)$, we find that for almost all $\boldsymbol{A}$ the matrix product is invertible, and a solution $\boldsymbol{\lambda}^i$ exists and thus a suitable $\boldsymbol{w}^i$ exists. To see this, we can assume that the basis $\boldsymbol{v}^{i,\cdot}$ is an orthonormal basis and expand $\boldsymbol{a}_i$ in this basis (and an irrelevant orthogonal complement). We conclude that for almost all $\boldsymbol{A}$ such a $\boldsymbol{w}^i$ exists. Since the union of null-sets is a null-set the same statement holds for almost all $\boldsymbol{A}$ for all $i$ at the same time and therefore we find a matrix $\boldsymbol{M}$ such that $\boldsymbol{A} \boldsymbol{M} = \mathrm{Id}_{d_z}$ and, moreover, $(\boldsymbol{w}^i)^\top \boldsymbol{B}_s \boldsymbol{w}^j = 0$ because this holds for all $\boldsymbol{w}^i \in V_i$ and $\boldsymbol{w}^j \in V_j$. $\qquad \square$

We now construct the diagonal matrices that will later correspond to $D^2 \boldsymbol{f}_s$.

**Lemma A.7.** *Assume $d_x \geq d_z$ Given $\boldsymbol{A} \in \mathbb{R}^{d_z \times d_x}$ of maximal rank and diagonal matrices $\boldsymbol{D}^1, \ldots, \boldsymbol{D}^{d_z} \in \mathbb{R}^{d_z \times d_z}$ we can find diagonal matrices $\boldsymbol{\Lambda}^1, \ldots, \boldsymbol{\Lambda}^{d_x} \in \mathbb{R}^{d_z \times d_z}$ such that for all $s \in [d_z]$*

$$\boldsymbol{D}^s = -\sum_{i=1}^{d_x} \boldsymbol{A}_{s,i} \boldsymbol{\Lambda}^i. \tag{A.18}$$

*Proof.* The proof is straightforward as soon as one observes that this is a linear equation for the diagonal entries of $\boldsymbol{\Lambda}^i$. Indeed, denoting by $\boldsymbol{\lambda} = (\boldsymbol{\Lambda}_{11}^1, \ldots, \boldsymbol{\Lambda}_{d_z,d_z}^1, \ldots, (\boldsymbol{\Lambda}_{d_z,d_z}^{d_x})^\top \in \mathbb{R}^{d_z \cdot d_x}$ the vector containing all diagonal entries of the matrices $\boldsymbol{\Lambda}^i$ and similarly $\boldsymbol{d} = (\boldsymbol{D}_{11}^1, \ldots, \boldsymbol{D}_{d_z,d_z}^1, \ldots, \boldsymbol{D}_{d_z,d_z}^{d_z})^\top \in \mathbb{R}^{d_z^2}$ for the diagonal entries of $\boldsymbol{D}^s$. Then we can rewrite equation A.18 as follows using the Kronecker product $\otimes$

$$(\boldsymbol{A} \otimes \mathrm{Id}_{d_z}) \boldsymbol{\lambda} = -\boldsymbol{d}. \tag{A.19}$$

Now the rank of the matrix $\boldsymbol{A} \otimes \mathrm{Id}_{d_z}$ is the product of the ranks, i.e., $d_z \min(d_x, d_z) = d_z^2 \leq d_x d_z$ and thus a solution $\boldsymbol{\lambda}$ exists. $\qquad \square$

With these technical lemmas at hand, we can prove the theorem which we now restate for convenience of the reader.

**Theorem 3.2.** *Assume that $d_x \geq d_z^3$. Let $\boldsymbol{B}_l \in \mathbb{R}^{d_x \times d_x}$ be symmetric matrices for $1 \leq l \leq d_z$. Then there is for any $\boldsymbol{x}_0 \in \mathbb{R}^{d_x}$ and for almost every $\boldsymbol{A} \in \mathbb{R}^{d_z \times d_x}$ a generator $\boldsymbol{f} \in \mathcal{F}_{\mathrm{int}}$ with a (left)-inverse $\boldsymbol{g} \in \mathcal{G}_{\mathrm{int}}$, such that $\boldsymbol{f}(0) = \boldsymbol{x}_0$ and $D\boldsymbol{g}(\boldsymbol{x}_0) = \boldsymbol{A}$ and $D^2 \boldsymbol{g}_l(\boldsymbol{x}_0) = \boldsymbol{B}_l$ for $1 \leq l \leq d_z$.*

*Proof of Theorem 3.2.* Clearly we can assume that $\boldsymbol{x}_0 = 0$. The key idea is that if we can ensure that equation A.2 holds for $\boldsymbol{z} = 0$ we can extend $\boldsymbol{f}$ and $\boldsymbol{g}$ such that $\boldsymbol{g} \circ \boldsymbol{f}(\boldsymbol{z}) = \boldsymbol{z}$ and $\boldsymbol{f} \in \mathcal{F}_{\mathrm{add}}$. To achieve this, we first apply Lemma A.5 and then find a matrix $\boldsymbol{M}$ such that $\boldsymbol{A} \boldsymbol{M} = \mathrm{Id}_{d_z}$ and $\boldsymbol{M}^\top \boldsymbol{B}_s \boldsymbol{M} \in \mathrm{Diag}(d_z)$. Then we apply Lemma A.7 and find matrices $\boldsymbol{\Lambda}^i$ such that

$$\boldsymbol{M}^\top \boldsymbol{B}_s \boldsymbol{M} + \sum_{i=1}^{d_x} \boldsymbol{A}_{s,i} \boldsymbol{\Lambda}^i = 0. \tag{A.20}$$

Now we pick a function $\boldsymbol{f} \in \mathcal{F}_{\mathrm{add}}$ such that $\boldsymbol{f}(0) = 0$, $D\boldsymbol{f}(0) = \boldsymbol{M}$ and $D^2 \boldsymbol{f}_i = \boldsymbol{\Lambda}^i$. Clearly, this is possible because $\boldsymbol{\Lambda}^i$ are diagonal, e.g., we can locally use a quadratic polynomial to achieve

this. The next step is to construct a function $\boldsymbol{g}$ such that $\boldsymbol{g} \circ \boldsymbol{f}(\boldsymbol{z}) = \boldsymbol{z}$. Using standard techniques (partition of unity) it is sufficient to construct this locally and then extend it globally. First, we consider $\bar{\phi} : \Omega \subset \mathbb{R}^{d_x} \times \mathbb{R}^{d_x}$ such that $\bar{\phi}(\boldsymbol{f}(\boldsymbol{z})) = (\boldsymbol{z}, 0)$ for $\boldsymbol{z} \in \Omega$ (e.g., by the existence of tubular neighbourhoods). We call the composition of $\bar{\phi}$ with the projection on the first $d_z$ components $\phi$. Then we define

$$\boldsymbol{g}(\boldsymbol{x}) = \boldsymbol{A}\boldsymbol{x} + \frac{1}{2} \begin{pmatrix} \boldsymbol{x}^\top \boldsymbol{B}_1 \boldsymbol{x} \\ \vdots \\ \boldsymbol{x}^\top \boldsymbol{B}_{d_z} \boldsymbol{x} \end{pmatrix} + \boldsymbol{h}(\phi(\boldsymbol{x})) \tag{A.21}$$

where $\boldsymbol{h}$ is given by

$$\boldsymbol{h}(\boldsymbol{z}) = \boldsymbol{h}(\phi(\boldsymbol{f}(\boldsymbol{z}))) = \boldsymbol{z} - \boldsymbol{A}\boldsymbol{f}(\boldsymbol{z}) - \frac{1}{2} \begin{pmatrix} \boldsymbol{f}(\boldsymbol{z})^\top \boldsymbol{B}_1 \boldsymbol{f}(\boldsymbol{z}) \\ \vdots \\ \boldsymbol{f}(\boldsymbol{z})^\top \boldsymbol{B}_{d_z} \boldsymbol{f}(\boldsymbol{z}). \end{pmatrix} \tag{A.22}$$

We can calculate

$$\boldsymbol{g}(\boldsymbol{f}(\boldsymbol{z})) = \boldsymbol{z} \tag{A.23}$$

so $\boldsymbol{g}$ is indeed a left-inverse of $\boldsymbol{f}$. Taking the derivative of this equation at 0 we obtain

$$D\boldsymbol{h}(0) = \mathrm{Id}_{d_z} - \boldsymbol{A}(D\boldsymbol{f}(0)) + 0 = \mathrm{Id}_{d_z} - \boldsymbol{A}\boldsymbol{M} = 0. \tag{A.24}$$

For the second derivative we get

$$D^2 \boldsymbol{h}_s(0) = -\sum_{i=1}^{d_x} \boldsymbol{A}_{s,i} D^2 \boldsymbol{f}_i(\boldsymbol{z}) - D\boldsymbol{f}(0)^\top \boldsymbol{B}_s D\boldsymbol{f}(0) = -\sum_{i=1}^{d_x} \boldsymbol{A}_{s,i} \boldsymbol{\Lambda}^i + \boldsymbol{M}^\top \boldsymbol{B}_s \boldsymbol{M} = 0. \tag{A.25}$$

Here we used for the derivative of the quadratic term that the contribution where the derivative hits one $\boldsymbol{f}(\boldsymbol{z})$ twice vanishes since $\boldsymbol{f}(0) = 0$. Finally, we can now evaluate

$$D\boldsymbol{g}(0) = \boldsymbol{A} + D\boldsymbol{h}(\phi(0)) = \boldsymbol{A} + D\boldsymbol{h}(0) = \boldsymbol{A} \tag{A.26}$$

and

$$D^2 \boldsymbol{g}_s(0) = \boldsymbol{B}_s + D^2 \boldsymbol{h}_s \circ \phi = \boldsymbol{B}_s \tag{A.27}$$

where $D^2 \boldsymbol{h}_s \circ \phi = 0$ follows from the chain rule and $D\boldsymbol{h}(0) = 0$ and $D^2 \boldsymbol{h}(0) = 0$. $\qquad \square$

## A.2 CONSTRAINING $\mathcal{G}_{\mathrm{enc}}$ BY ARCHITECTURE

In this section we discuss results showing that it is challenging to construct practical function classes $\mathcal{G}_{\mathrm{enc}}$ which are sufficiently expressive so that they contain a left-inverse for each $\boldsymbol{f} \in \mathcal{F}_{\mathrm{int}}$. As explained before, the main challenge is that setting $\mathcal{G}_{\mathrm{enc}} = \mathcal{G}_{\mathrm{int}}$ is in principle sufficient to ensure identifiability and out of distribution generalization. So we need to make additional assumptions on $\mathcal{G}_{\mathrm{enc}}$ that function classes used in widely applied algorithms satisfy which then ensure that $\mathcal{G}_{\mathrm{enc}}$ is very expressive preventing that equation 2.6 holds. We will make the assumption that $\mathcal{G}_{\mathrm{enc}}$ has a linear structure, i.e., $\boldsymbol{g}_1 + \boldsymbol{g}_2 \in \mathcal{G}_{\mathrm{enc}}$ if $\boldsymbol{g}_1, \boldsymbol{g}_2 \in \mathcal{G}_{\mathrm{enc}}$. This is clearly satisfied when $\mathcal{G}_{\mathrm{enc}}$ is a vector space (e.g., this assumption is satisfied for linear or kernel methods, or when learning a linear head on a fixed representation). For functions implemented by neural networks with fixed architecture this is in general not true. However, it does apply to infinite width limits of fixed architectures (this does not generally imply universal approximation properties when the architecture is sparse, e.g., we use slot-wise neural networks for the forward direction which cannot approximate interactions $\boldsymbol{x}_1 \boldsymbol{x}_2$ even at infinite width). Note that large width is also generally required to make neural networks sufficiently expressive because for fixed width neural networks implement a parametric function class while $\mathcal{F}_{\mathrm{int}}$ is non-parametric. We then show that such a function class $\mathcal{G}_{\mathrm{enc}}$ does not have a useful inductive bias.

**Architecture constraints for $d_x = d_z$.** We first consider the simpler case $d_x = d_z$ where $\boldsymbol{f}$ is bijective on the codomain (and not only on its image).

Our main result here is that $\mathcal{G}_{\text{enc}}$ has the universal approximation property when $\mathcal{G}_{\text{enc}} \supset \mathcal{F}_{\text{add}}^{-1}$ and $\mathcal{G}_{\text{enc}}$ is closed under addition.

**Theorem A.8.** *Assume $d_x = d_z = d$. Consider an encoder function class $\mathcal{G}_{\text{enc}}$ with the following two properties:*

1. *The class $\mathcal{G}_{\text{enc}}$ is closed under addition, i.e., for $\boldsymbol{g}_1, \boldsymbol{g}_2 \in \mathcal{G}_{\text{enc}}$ also $\boldsymbol{g}_1 + \boldsymbol{g}_2 \in \mathcal{G}_{\text{enc}}$.*

2. *The function class $\mathcal{G}_{\text{enc}}$ is expressive enough such that it contains all inverses of additive functions, i.e., $\mathcal{F}_{\text{add}}^{-1} \subset \mathcal{G}_{\text{enc}}$.*

*Then $\mathcal{G}_{\text{enc}}$ is dense in the space of all continuous functions on all compact subset of $\mathbb{R}^{d_x}$.*

Since $\mathcal{F}_{\text{add}} \subset \mathcal{F}_{\text{int}}$ for $n \geq 1$ and any $m$ we directly get the following corollary.

**Corollary A.9.** *Assume $d_x = d_z = d$ and the encoder function class $\mathcal{G}_{\text{enc}}$ is closed under addition and satisfies $\mathcal{G}_{\text{int}} \subset \mathcal{G}_{\text{enc}}$. Then $\mathcal{G}_{\text{enc}}$ is dense in the space of all continuous functions on all compact subset of $\mathbb{R}^{d_x}$.*

The takeaway from these results is that it is challenging to find natural function classes $\mathcal{G}_{\text{enc}}$ so that $\mathcal{G}_{\text{enc}} \supset \mathcal{G}_{\text{int}}$ (sufficient expressivity) but $\mathcal{G}_{\text{enc}}$ is not much larger than $\mathcal{G}_{\text{int}}$. Therefore, learning only encoders from $\mathcal{G}_{\text{enc}}$ does not provide a strong inductive bias towards the inverse of the ground truth decoder and out of distribution generalization.

*Proof of Theorem A.8.* The general strategy is to prove that the conditions imply that all maps $\boldsymbol{g}$ where $\boldsymbol{g}_j$ (the $j$-th coordinate of $\boldsymbol{g}$) is any polynomial and $\boldsymbol{g}_i = 0$ for $i \neq j$ are contained in $\mathcal{G}_{\text{enc}}$. This will end the proof because polynomials are dense in the scalar valued continuous functions, and we can then apply this result coordinate-wise using the additive structure.

**Step 1:** Vector space structure of $\mathcal{G}_{\text{enc}}$. We now show that we can scale certain functions in $\mathcal{G}_{\text{enc}}$. Denote for $\lambda \neq 0$ by $\boldsymbol{M}_\lambda$ the multiplication map $z \to \lambda z$. Then $\boldsymbol{f} \circ \boldsymbol{M}_\lambda \in \mathcal{F}_{\text{add}}$ if $\boldsymbol{f} \in \mathcal{F}_{\text{add}}$. Since $(\boldsymbol{f} \circ \boldsymbol{M}_\lambda)^{-1} = \lambda^{-1} \boldsymbol{f}^{-1}$ we conclude that scalar multiples of $\boldsymbol{f}^{-1}$ are in $\mathcal{F}_{\text{add}}^{-1}$ and the first assumption then implies that the vector space $V$ generated by $\boldsymbol{f}^{-1}$ for $\boldsymbol{f} \in \mathcal{F}_{\text{add}}$ is contained in $\mathcal{G}_{\text{enc}}$.

**Step 2:** We show that the monomials $x_i^k$ are contained in $\mathcal{G}_{\text{enc}}$. Consider the map $\boldsymbol{f} \in \mathcal{F}_{\text{add}}$ where $\boldsymbol{x} = \boldsymbol{f}(\boldsymbol{z})$ has coordinates

$$\begin{aligned} \boldsymbol{x}_1 &= \boldsymbol{z}_2^k + \boldsymbol{z}_1 \\ \boldsymbol{x}_i &= \boldsymbol{z}_i \quad \text{for } d \geq i \geq 2. \end{aligned} \tag{A.28}$$

This is clearly an additive function with inverse

$$\begin{aligned} \boldsymbol{z}_1 &= \boldsymbol{x}_1 - \boldsymbol{x}_2^k \\ \boldsymbol{z}_i &= \boldsymbol{x}_i \quad \text{for } d \geq i \geq 2. \end{aligned} \tag{A.29}$$

Similarly, we consider

$$\begin{aligned} \boldsymbol{x}_1 &= -(-\boldsymbol{z}_2)^k - \boldsymbol{z}_1 \\ \boldsymbol{x}_i &= -\boldsymbol{z}_i \quad \text{for } d \geq i \geq 2. \end{aligned} \tag{A.30}$$

with inverse

$$\begin{aligned} \boldsymbol{z}_1 &= -\boldsymbol{x}_1 - \boldsymbol{x}_2^k \\ \boldsymbol{z}_i &= -\boldsymbol{x}_i \quad \text{for } d \geq i \geq 2. \end{aligned} \tag{A.31}$$

Summing these two functions, we find that the function $\boldsymbol{g}$ with

$$\boldsymbol{g}_i(x) = -2\delta_{1i}\boldsymbol{x}_1^k \tag{A.32}$$

satisfies $\boldsymbol{g} \in \mathcal{G}_{\text{enc}}$. By permutation of the outputs and inputs (and scaling) we find that all functions $\boldsymbol{g}$ with $\boldsymbol{g}_i(\boldsymbol{x}) = \delta_{ij}\boldsymbol{x}_l^k$ are in $\mathcal{G}_{\text{enc}}$ for all $j, l \in [d]$ and $k \in \mathbb{N}$.

**Step 3:** Now we show with a similar argument that more generally functions of the form $g_j(x) = \delta_{jl}(\sum_{i=1}^{d} \alpha_i x_i)^k$ for all coefficients $\alpha_i$ and all $1 \leq l \leq d$ are in $\mathcal{G}_{\mathrm{enc}}$. If only one $\alpha_i$ is non-zero we have shown this before, so we can assume that at least two $\alpha_i$ are non-zero and without loss of generality we assume that $\alpha_i$ for $1 \leq i \leq k$ are non-zero where $2 \leq k \leq d$. Then we consider the additive map $g$ which satisfies for $x = g(z)$

$$
\begin{aligned}
x_1 &= \frac{1}{\alpha_1}\left(z_1^k + z_1 - \sum_{i=2}^{k} z_i\right), \\
x_2 &= \frac{1}{\alpha_2}(z_2 - z_1^k), \\
x_i &= \frac{1}{\alpha_i} z_i \qquad \text{for } 3 \leq i \leq k, \\
x_i &= z_i \quad \text{for } k < i \leq d.
\end{aligned}
\tag{A.33}
$$

Then we observe that

$$
\sum_{i=1}^{k} \alpha_i x_i = z_1
\tag{A.34}
$$

and thus the inverse is given by

$$
\begin{aligned}
z_1 &= \sum_{i=1}^{k} \alpha_i x_i, \\
z_2 &= \alpha_2 x_2 + \left(\sum_{i=1}^{k} \alpha_i x_i\right)^k, \\
z_i &= \alpha_i x_i \quad \text{for } 3 \leq i \leq k \\
z_i &= x_i \qquad \text{for } d \geq i > k.
\end{aligned}
\tag{A.35}
$$

Similarly, we find that the inverse of the additive map given in coordinates by

$$
\begin{aligned}
x_1 &= \frac{1}{\alpha_1}\left(-(-z_1)^k - z_1 + \sum_{i=2}^{k} z_i\right), \\
x_2 &= \frac{1}{\alpha_2}(-z_2 + (-z_1)^k), \\
x_i &= -\frac{1}{\alpha_i} z_i \qquad \text{for } 3 \leq i \leq k, \\
x_i &= z_i \quad \text{for } k < i \leq d.
\end{aligned}
\tag{A.36}
$$

can be written as (note $\sum_{i=1}^{k} \alpha_i x_i = -z_1$)

$$
\begin{aligned}
z_1 &= -\sum_{i=1}^{k} \alpha_i x_i, \\
z_2 &= -\alpha_2 x_2 + \left(\sum_{i=1}^{k} \alpha_i x_i\right)^k, \\
z_i &= -\alpha_i x_i \quad \text{for } 3 \leq i \leq k \\
z_i &= -z_i \qquad \text{for } d \geq i > k.
\end{aligned}
\tag{A.37}
$$

Summing the two inverses in equation A.33 and equation A.37 we find that the map $g$ given by $g_j(x) = 2\delta_{j2}\left(\sum_{i=1}^{k} \alpha_i x_i\right)^k$ is in $\mathcal{G}_{\mathrm{enc}}$ and by permuting the indices and scaling we find that all maps of the form

$$
g_j(x) = \delta_{jl}\left(\sum_{i=1}^{k} \alpha_i x_i\right)^k
\tag{A.38}
$$

are in $\mathcal{G}_{\text{enc}}$. Using Lemma A.10 stated below we infer that indeed all multinomial polynomials are in $\mathcal{G}_{\text{enc}}$ and this ends the proof in light of the Stone-Weierstrass Theorem. $\qquad\square$

The following technical but standard lemma was used in the proof of Theorem A.8.

**Lemma A.10.** *Consider the space of functions* $\boldsymbol{g_\alpha} : \mathbb{R}^d \to \mathbb{R}$ *for* $\boldsymbol{\alpha} \in \mathbb{R}^d$ *given by*

$$\boldsymbol{g}_\alpha(\boldsymbol{x}) = \left( \sum_{i=1}^{d} \boldsymbol{\alpha}_i \boldsymbol{x}_i \right)^k. \tag{A.39}$$

*Then the vector space generated by the functions* $\boldsymbol{g_\alpha}$ *is the space of all $k$-homogeneous polynomials.*

*Proof.* This is a general version of the well known polarization identity, namely

$$(\boldsymbol{x}_1 + \boldsymbol{x}_2)^2 - (\boldsymbol{x}_1 - \boldsymbol{x}_2)^2 = 4\boldsymbol{x}_1\boldsymbol{x}_2. \tag{A.40}$$

For completeness we sketch the full proof. Denote the generated space by $V$. Let $\phi_j(\boldsymbol{x})$ be linear functions for $1 \leq j \leq k$, i.e., $\phi_j(\boldsymbol{x}) = \sum_{i=1}^{d} \boldsymbol{\alpha}_i^j \boldsymbol{x}_i$ for some $\boldsymbol{\alpha}_i^j$. Then using the multnomial expansion we find

$$\sum_{(\epsilon_1,\dots,\epsilon_k)\in\{-1,1\}^k} \left( \prod_{j=1}^{k} \epsilon_j \right) \left( \sum_{j=1}^{k} \epsilon_j \phi_j \right)^k$$

$$= \sum_{(\epsilon_1,\dots,\epsilon_k)\in\{-1,1\}^k} \left( \prod_{j=1}^{k} \epsilon_j \right) \sum_{\gamma_1+\dots+\gamma_k=k} \frac{k!}{\gamma_1! \cdot \dots \cdot \gamma_k!} \prod_{i=j}^{k} \phi_j^{\gamma_j}$$

$$= \sum_{(\epsilon_1,\dots,\epsilon_k)\in\{-1,1\}^k} \left( \prod_{i=j}^{k} \epsilon_j \right) \sum_{\gamma_1+\dots+\gamma_k=k} \frac{k!}{\gamma_1! \cdot \dots \cdot \gamma_k!} \prod_{j=1}^{k} (\epsilon_j \phi_j)^{\gamma_j}$$

$$= \sum_{\gamma_1+\dots+\gamma_k=k} \frac{k!}{\gamma_1! \cdot \dots \cdot \gamma_k!} \prod_{j=1}^{k} \phi_j^{\gamma_j} \prod_{j=1}^{k} \sum_{\epsilon_j\in\{-1,1\}} \epsilon_j^{\gamma_j+1}. \tag{A.41}$$

Now the last sum is 0 for $\gamma_j$ even and 2 for $\gamma_j$ odd. So the only non-zero term corresponds to all $\gamma_j$ odd and thus $\gamma_j = 1$ for all $j$ and therefore

$$\sum_{(\epsilon_1,\dots,\epsilon_k)\in\{-1,1\}^k} \left( \prod_{j=1}^{k} \epsilon_j \right) \left( \sum_{j=1}^{k} \epsilon_j \boldsymbol{\alpha}_j \right)^k = 2^k k! \prod_{j=1}^{k} \phi_j \in V. \tag{A.42}$$

Clearly this implies that the monomials $\prod_{i=1}^{d} \boldsymbol{x}_i^{\boldsymbol{\beta}_i}$ with $\boldsymbol{\beta}_i \geq 0$ and $\sum_{i=1}^{d} \boldsymbol{\beta}_i = k$ are generated by the functions $\boldsymbol{g_\alpha}$ (pick $\phi_j(\boldsymbol{x}) = \boldsymbol{x}_i$ for $\boldsymbol{\beta}_i$ of the $\phi_j$). $\qquad\square$

**Architectural constraints for** $d_x > d_z$**.** The case $d_x > d_z$ is more challenging because the data manifold is then a submanifold and even if we know that there is a $\boldsymbol{g} \in \mathcal{G}_{\text{enc}}$ inverting $\boldsymbol{f}$ on the data manifold (i.e., $\mathcal{G}_{\text{enc}}$ is sufficiently expressive) this provides (essentially) no information about $\boldsymbol{g}$ away from $\mathcal{X} = \boldsymbol{f}(\mathcal{Z})$ and the data manifolds for different generators are unrelated. However, we can leverage the result for $d_x = d_z$ to obtain a weaker version in the general case. Here we make the additional assumption that $\mathcal{G}_{\text{enc}}$ is closed under coordinate projections in the sense that $\tilde{\boldsymbol{g}} \in \mathcal{G}_{\text{enc}}$ if $\tilde{\boldsymbol{g}}(\boldsymbol{x}) = \boldsymbol{g}(\boldsymbol{x}_I, \boldsymbol{0}_{[d_x]\setminus I})$ for some index set $I \subset [d_x]$ and $\boldsymbol{g} \in \mathcal{G}_{\text{enc}}$. Note that this is naturally satisfied for neural networks where we can remove the influence of a coordinate by zeroing its outgoing weights.

**Corollary A.11.** *Assume that* $\mathcal{G}_{\text{enc}}$ *is a class of encoder functions such that* $\mathcal{G}_{\text{enc}}$ *is closed under addition and coordinate projections and sufficiently expressive, i.e., for every* $\boldsymbol{f} \in \mathcal{F}_{\text{int}}$ *there is* $\boldsymbol{g} \in \mathcal{G}_{\text{enc}}$ *such that* $\boldsymbol{g} \circ \boldsymbol{f} = \text{id}$. *Let* $\boldsymbol{f} \in \mathcal{F}_{\text{int}}$ *be such that* $\boldsymbol{f}_I$ *is a diffeomorphism (on its image) for some* $I$ *with* $|I| = d_z$. *Then* $\mathcal{G}_{\text{enc}} \circ \boldsymbol{f}$ *is dense in all continuous functions* $C(K, \mathbb{R}^{d_z})$ *for every compact* $K \subset \mathbb{R}^{d_z}$, *i.e., essentially arbitrary representations can be learned using function in* $\mathcal{G}_{\text{enc}}$.

*Proof.* Consider a set $I \subset [d_z]$. Then the restrictions $\boldsymbol{f}_I$ of functions $\boldsymbol{f} \in \mathcal{F}_{\text{int}}$ such that $\boldsymbol{f}_{I^c}(\boldsymbol{z}) = \boldsymbol{0}$ (i.e., functions that vanish in all but $d_z$ coordinates) are in bijection to functions in $\mathcal{F}_{\text{int}}$ mapping $\mathbb{R}^{d_z} \to \mathbb{R}^{d_z}$. Applying Theorem A.8 we therefore find that the set of functions $\boldsymbol{z}_I \to \boldsymbol{g}(\boldsymbol{z}_I, \boldsymbol{0}_{I^c})$ for $\boldsymbol{g} \in \mathcal{G}_{\text{enc}})$ is dense in the continuous functions defined on any compact set $K'$. It is convenient to introduce the shorthand $\bar{\boldsymbol{g}}$ for the function $\boldsymbol{z}_I \to \boldsymbol{g}(\boldsymbol{z}_I, \boldsymbol{0}_{I^c})$ by $\bar{\boldsymbol{g}}$. Then we can restate the density statement before as follows: Given any continuous function $\boldsymbol{h} : K \to \mathbb{R}^{d_z}$ we can find for any $\epsilon > 0$ a $\boldsymbol{g} \in \mathcal{G}_{\text{enc}}$ so that $\|\bar{\boldsymbol{g}} - \boldsymbol{h} \circ (\boldsymbol{f}_I)^{-1}\| < \epsilon$ on the compact set $K' = \boldsymbol{f}_I(K)$ (here we use that $\boldsymbol{f}_I$ is bijective on its image to invert it). Using that $\mathcal{G}_{\text{enc}}$ is closed under coordinate projections we can find $\tilde{\boldsymbol{g}}$ is in $\mathcal{G}_{\text{enc}}$ and satisfies

$$
\begin{aligned}
\max_{\boldsymbol{z} \in K} \|\tilde{\boldsymbol{g}}\boldsymbol{f}(\boldsymbol{z}) - \boldsymbol{h}(\boldsymbol{z})\|_\infty &= \|\boldsymbol{g}(\boldsymbol{f}_I(\boldsymbol{z}), \boldsymbol{0}_{I^c}) - \boldsymbol{h} \circ (\boldsymbol{f}_I)^{-1} \circ \boldsymbol{f}_I(\boldsymbol{z})\|_\infty \\
&= \|\bar{\boldsymbol{g}}(\boldsymbol{f}_I(\boldsymbol{z})) - \boldsymbol{h} \circ (\boldsymbol{f}_I)^{-1}(\boldsymbol{f}_I(\boldsymbol{z}))\|_\infty \qquad\text{(A.43)} \\
&\leq \max_{\boldsymbol{x}_I \in K'} \|\bar{\boldsymbol{g}}(\boldsymbol{x}_I) - \boldsymbol{h} \circ (\boldsymbol{f}_I)^{-1}(\boldsymbol{x}_I)\|_\infty.
\end{aligned}
$$

This ends the proof.

$\square$

While the previous corollary makes the strong assumption that $\boldsymbol{f}_I$ is globally bijective we note that this is generally true at least locally. Moreover, we can patch such representations due to the additive structure. Therefore, it seems unlikely that a function class $\mathcal{G}_{\text{enc}}$ satisfying the following three constraints exists: First, the function class $\mathcal{G}_{\text{enc}}$ is expressive enough, i.e., it contains left inverses for all $\boldsymbol{f} \in \mathcal{F}_{\text{int}}$. Secondly, $\mathcal{G}_{\text{enc}}$ is not too expressive so it provides a useful inductive bias towards $\mathcal{G}_{\text{int}}$. And finally, $\mathcal{G}_{\text{enc}}$ can be efficiently parametrized and used for optimization.

# B EXPERIMENTAL DETAILS

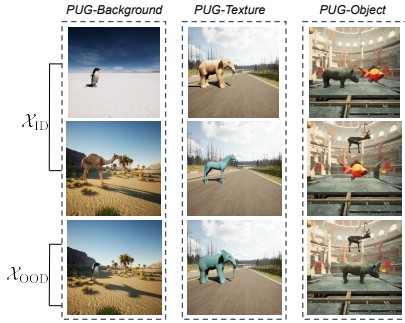 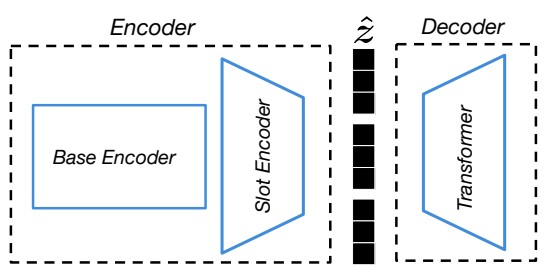

Figure 7: Left. Overview of the data splits used in the experiments. PUG-Background contains unseen combinations of background and object in its OOD split $\mathcal{X}_{\text{OOD}}$, PUG-Texture contains unseen object-texture combinations in $\mathcal{X}_{\text{OOD}}$, and PUG-Object contains unseen object combinations in $\mathcal{X}_{\text{OOD}}$. Right. General structure of the employed models. A *base encoder* (pretrained for most experiments) is used to extract features from the images which are then mapped through a *slot encoder* which leverages a cross-attention mechanism and potentially self-attention. For our decoders we use the regularized cross-attention Transformer architecture from Brady et al. (2025)
.

## B.1 DATA

We create datasets for our experiments in Sec. 5 based on the PUG: Animals dataset (Bordes et al., 2023). This data consists of 43,520 high-resolution images which we resize to $224 \times 224 \times 3$. To create PUG-Background, we create an OOD set containing 32,000 images which consist of unseen combinations of animal category and backgrounds, e.g., penguin in a desert in Fig. 7, and a corresponding ID set containing 11,520 images. For PUG-Texture, the OOD set contains 16,000 images consisting of unseen combinations of animal and texture/color, e.g. blue elephant in Fig. 7, and the ID set contains 27,520 images. Lastly, for PUG-Object, the ID and OOD set both contain 21,760

images. The OOD set here consist of unseen combinations of animal categories, e.g., rhinoceros and caribou in in Fig. 7.

## B.2 MODELS

**Base encoders.** We use six different pretrained base encoders along with a ViT small for our experiments in Sec. 5. The specific sizes of each model can be seen in Fig. 5. When fine tuning these models with a LORA adapter, we use a rank of 16, a scaling factor of 32, and a dropout value of 0.1.

**Slot encoders.** We use either a Transformer or Slot Attention model for the slot encoder in Sec. 5. The transformer model consist of both self and cross-attention layers. Both models consist of 5 layers. We use 3 slots for each model, with dimensions of 64.

**Decoders.** All decoders in our experiments use the cross-attention Transformer from Brady et al. (2025); Jaegle et al. (2022); Sajjadi et al. (2022). In this model, slots are first projected by a 2 layer slot-wise MLP and then passed through 2 layers of a cross-attention Transformer with pixel queries. Pixels are tokenized using a 2 layer MLP. The pixel outputs of the cross-attention Transformer are mapped to have a channel dimension of 3 using a 3 layer MLP. The attention weights in this model are regularized using the regularizer introduced by Brady et al. (2025). For all experiments, we use a value of 0.01 for this loss.

**Readout.** We use a single layer linear readout shared across slots to predict animal or background categories from slots.

## B.3 TRAINING OBJECTIVES

**Supervised models.** We train all supervised models for 100000 iterations across 3 random seeds using a batch size of 64, with the Adam optimizer (Kingma and Ba, 2015) and a learning rate of 1e-4.

**VAEs.** We train all unsupervised VAE models for 300000 iterations across 3 random seeds using a batch size of 32, with the Adam optimizer (Kingma and Ba, 2015) and a learning rate of 5e-4, which is decayed by a factor of .1 throughout training and warmed up for the first 10000 iterations. We use a value of either 0.005 or 0.001 for the hyperparameter $\beta$ on the KL loss.

**Readout.** In our unsupervised experiments, we train a linear readout on learned slots for 7500 iterations. To resolve the permutation between inferred and ground-truth slots we rely on on the Hungarian matching procedure used in Dittadi et al. (2022); Locatello et al. (2020b).

**Gradient-based search.** When performing gradient based search in our experiments, we optimize Eq. 4.3 using Adam with a learning rate of .001. We optimize for either 300 or 500 iterations on PUG-Background and 700 iterations on PUG-Texture. To further aid in optimization we add an additional regularizer to the optimization procedure which minimizes the entropy of the logits under the classifier. This aims to ensure that the search procedure yields latent slots which are within the set of slots which the classifier has already observed. We use a value of either 10 or 50 for this loss. We note that a similar loss was used for semi-supervised learning in Grandvalet and Bengio (2004).

**Generative replay.** For our experiments using generative replay, we generate OOD data by following the procedure in Wiedemer et al. (2024a) in which ID slots are randomly shuffled to create novel compositions. We train an encoder on batches of 64 of OOD samples for 15000 iterations with a learning rate of 5e-4.

**Compute.** We train all models using 2 NVIDIA A100 GPUs. Total training time was approximately 1500h.

```python
import torch
from torch.optim import Adam

def latent_search(x_ood, encoder, decoder, learning_rate, num_iters):

    # get encoding for ood image
    zh = encoder(x_ood)
    optimizer = Adam(params=[zh],
    lr=learning_rate)

    # optimize encoding to minimize mse under decoder
    for _ in range(num_iters):
        xh = decoder(zh)
        mse_loss = (x_ood - xh).square().mean()
        mse_loss.backward()
        optimizer.step()

    return zh

def generative_replay(zh_ood, encoder, decoder, learning_rate):

    optimizer = Adam(params=encoder.parameters(),
    lr=learning_rate)

    # ensure gradients not computed for decoder
    for param in decoder.parameters():
        param.requires_grad = False

    # generate OOD image
    xh_ood = decoder(zh_ood)

    # re-encode image with encoder
    zh_ood_recon = encoder(xh_ood)

    # ensure re-encoding matches original encoding
    mse_loss = (zh_ood - zh_ood_recon).square().mean()
    mse_loss.backward()
    optimizer.step()

    return encoder
```

Figure 8: Top. PyTorch pseudocode for gradient-based search using a decoder for a given OOD image $x$. Bottom. PyTorch pseudocode for one gradient step of generative replay.

## C    ADDITIONAL EXPERIMENTS

**Setup.**

Our experiments in § 5 highlight that generative methods can yield significant gains in compositional generalization. However, these results are only shown for one decoder architecture. Consequently, it remains unclear whether gains in OOD performance come from our specific setup, which leverages a structured slot-based decoder, or whether a less-structured decoder can yield similar gains. To test this, we train an autoencoder on PUG-Background with the following structure: images are mapped by a CNN encoder to a latent representation with the same total dimensionality as our slot-based model, but which does not have a slot structure. This latent is then decoded by an CNN decoder which is not slot-based. We train this model on PUG-Background across 3 seeds using the same training settings as § 5 and with a VAE loss with $\beta = 0.005$. For the CNN encoder and decoder, we leverage the same architecture used in Dittadi et al. (2022). To evaluate classification performance, we train 3 separate linear readouts on the latents of this model to predict each animals and background. To evaluate whether this decoder offers gains in OOD performance, we perform 750 iterations of search § 4.1 under the trained decoder and re-evaluate OOD accuracy.

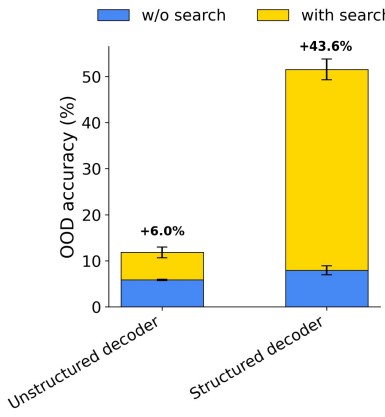

Figure 9: OOD accuracy on PUG-Background after applying search § 4.1 using a structured slot-based Transformer decoder designed according to § 3 vs. an unstructured CNN decoder.

We benchmark the performance of this unstructured model against our structured Transformer decoder from § 5 which uses a ViT-Small base encoder trained from scratch. Similarly, we perform 750 iterations of search under this decoder, and re-evaluate OOD performance.

**Results.**

For both models, we observe near-perfect ID classification performance. Thus, similar to § 5, we only report OOD performance. Results can be seen in Fig. 9. We find that both models achieve similar OOD accuracy before leveraging search. However, after applying search, we observe significantly higher gains in OOD performance when using the structured Transformer decoder compared to the unstructured CNN. These results suggest that non-trivial gains in compositional generalization using generative methods require constraining a decoder's structure as highlighted by our theory in § 3.

## D  EXTENDED DISCUSSION

**More complex datasets.** One limitation of our experimental study is that we do not evaluate compositional generalization on real-world image datasets. The main issue in leveraging such datasets is that evaluating compositional generalization requires data in which all ground-truth latent factors in each image (e.g. objects, backgrounds, textures, etc.) are explicitly known, thereby enabling us to construct ID and OOD splits in a principled way. For real-world, unstructured data the underlying latents are unknown making such datasets unsuitable for rigorously testing compositional generalization. Consequently, we elected to use PUG, which to the best of our knowledge is the most visually complex data in which the underlying latents are known. Yet, this dataset is still limited in its visual complexity relative to real world data. We thus believe that an important direction for future work is to create a large scale, visually realistic image dataset which offers access to the ground-truth latents, perhaps by leveraging a synthetic or neural network based image renderer.

**Compositional generalization via data scale.** A key result of our experiments in § 5 is that non-generative methods leveraging encoders with large-scale pretraining achieve substantial gains in OOD performance. This suggests that, as the pretraining size of the base encoder increases, the full encoder (base encoder plus slot encoder) becomes initialized in a region of the loss landscape where all reachable optima correspond to models that also generalize OOD. A limitation of our current experiments is that we cannot investigate this phenomenon in a more systematic manner, since we rely on pretrained encoders that differ in architecture, training objective, and pretraining dataset. A more principled approach would require an empirical study in which an encoder's architecture and training objective are held fixed, while the model is trained on gradually increasing amounts of pretraining data. We leave such an investigation as an important direction for future work.

**Computational cost of generative methods.** One drawback of leveraging generative methods is that training a decoder as well as using search or replay add computational overhead relative to encoder-only methods. For example, in our experiments in § 5.2, a single iteration for a supervised ViT-Small base encoder took $\sim 55$ ms for a batch size of 32, while adding a Transformer decoder and training with a VAE loss increased this time to $\sim 366$ ms. This overhead can be reduced by only decoding a subset of pixels similar to van Steenkiste et al. (2024), but we leave this for future work. Additionally, using generative replay incurs approximately $\sim 170$ ms per iteration, while search requires $\sim 215$ ms with a batch size of 32. We note that we train for only 15,000 replay iterations, making this procedure relatively computationally scalable. Search, on the other hand, requires 750 iterations per batch. Thus, further work is needed to scale this procedure more effectively.

**Limitations of assumptions in theory.** Our theoretical results in § 3, rely on two main assumptions on the ground-truth generator $f$: (i) that $f$ is a diffeomorphism and (ii) that the interactions between slots under $f$ are restricted according to Eq. (2.7). While these assumptions are the most general which have been shown to enable compositional generalization, it is possible that they may fail to accurately model real-world data in certain circumstances. For example, the assumption that $f$ is a diffeomorphism may not hold for images in which objects exhibit strong occlusions such that $f$ is

no longer invertible. Additionally, it is possible that certain interactions between concepts may not be captured by Eq. (2.7) such as complex shadows or reflections.

**Limitations of assumptions in practice.** Enforcing our theoretical assumptions on a model in practice can potentially pose scalability challenges. For example, enforcing that a model is a diffeomorphism via an autoencoder is computationally costly and thus can be challenging to scale. We note, however, that modern generative diffusion models and classifiers (Jaini et al., 2024; Peebles and Xie, 2023; Rombach et al., 2022) rely on learning a diffeomorphic mapping between image and latent space, highlighting the feasibility of enforcing this assumption at scale. Furthermore, enforcing that a decoder takes the form of Eq. (2.7) exactly is not scalable in practice (Brady et al., 2025). To this end, we aim to approximate this form in our experiments using a structured Transformer decoder and restricting interactions via a KL penalty and a sparsity regularizer on the decoder's attention weights (Brady et al., 2025). Importantly, we find this model yields significant gains for compositional generalization highlighting that enforcing that a decoder exactly matches Eq. (2.7) may not be necessary in practice. Furthermore, we suspect that biologically plausible regularizers such as in Whittington et al. (2023) which aim to minimize activation energy through sparsity regularizers on the latents and the decoder's weight matrix could serve a similar role as our regularizers.

