# OpenReview forum: "Generation is Required for Data-Efficient Perception"
_ICLR.cc/2026/Conference — Submitted to ICLR 2026_

### Official Review · Reviewer_zHMs · 2025-10-17

**Soundness:** 3
**Presentation:** 3
**Contribution:** 3
**Rating:** 6
**Confidence:** 2

**Summary:**

(Preliminary for disambiguate: the encoder/decoder refers to functions that take an image to a latent vector and vice versa. These are not the encoder/decoder in a transformer.)

This paper investigates whether using an invertible latent->image decoder helps compositional generalization (A+B, C+D are in-distribution, but A+C is not). The authors define the inductive biases for compositional generalizing in both encoder and decoder-based methods and provide theoretical analysis on why enforcing constraints is only practical at the decoder side. The authors propose efficient inversion mechanisms, i.e., gradient-based search and generative replay. The authors also built a synthetic image dataset with different animals and background combinations to validate their claim. With limited data, generative methods that can generate or search in the latent space are able to classify better than non-generative models.

**Strengths:**

-	The paper theoretically shows the necessity for generative modeling and then uses a sample toy example to validate this idea.
-	The decoder inversion mechanisms of gradient-based search and replay are novel.
-	The experiment results align with the author’s claim that encoders are less generalizable.
-	The paper is organized in a clear way.

**Weaknesses:**

-	The abstract is a bit confusing: (1) some of today’s most successive machine vision models, like VLM, have generative decoders; (2) the difference between generative and non-generative is in the distribution that they model, not decoder vs encoder architecture (only from Section 2 we know the encoder-decoder are defined as latent-image transformation functions); (3) the claim that enforcing inductive bias on decoder is easier than in encoder is not a well-known thing and thus need some explanation. It is recommended to define the encoder/decoder early to disambiguate from the encoder/decoder in the popular transformer architectures.
-	The theoretical claims are based on the assumptions that (1) data is generated by a diffeomorphic function and (2) components in an image can be cut into distinct slots in the latent vector. This limits the use case of the model since real images/other datasets may not satisfy these.
-	The generative replay is essentially doing synthetic data augmentation. What if, for the feedforward baselines, we also add synthetic data to the training set?
-	It is recommended to add pseudo code for the gradient-based search and replay.

**Questions:**

-	Text in Fig 5/6 is too small.
-	Could the authors elaborate a bit or give an example when the requirements for Eq 2.5/2.6 are not satisfied?
-	The gradient-based search seems to depend on an encoder. How sensitive is it to the errors in the encoder?

---

> ### Author Response · Authors · 2025-11-21
> **Official Comment by Authors (1/2)**
>
> We thank the reviewer for their time and their positive assessments of our work. We also appreciate the constructive feedback given by the reviewer. We address each of your comments below:
>
> **Comment**:  “The abstract is a bit confusing… ...the difference between generative and non-generative is in the distribution that they model, not decoder vs encoder architecture … It is recommended to define the encoder/decoder early to disambiguate from the encoder/decoder in transformers.”
>
> **Response**:
>
> We thank the reviewer for highlighting this confusion and we agree that the clarity of our definitions of generative and non-generative can be improved. We have thus updated the first two sentences of the abstract to emphasize that (i) our encoders map from images to latents and decoders from latents to images and (ii) that the distinction between generative and non-generative that we use is in terms of whether or not a model inverts a decoder.
>
> Furthermore, we have updated the first paragraph of our introduction to better reflect this point. We have also rewritten lines 131-135 in Section 2 to communicate the distinction between generative and non-generative methods more clearly.
>
> **Comment**: “some of today’s most successive machine vision models, like VLMs, have generative decoders”
>
> **Response**:
>
> While many VLMs include decoders, we emphasize that these decoders generally map an image’s latent representation to an image caption and are thus language decoders. Our results highlight that compositional generalization in vision requires an image decoder mapping latent representations to images. To highlight this, we have updated the second sentence of the abstract to include the phrase “image decoder”. We hope that this change along with our prior changes of defining our encoders and decoders more explicitly now communicate this distinction clearly.
>
> **Comment**: “the claim that enforcing inductive bias on decoder is easier than in encoder is not a well-known thing and thus need some explanation”
>
> **Response**:
>
> We assume the reviewer is referring to the line in the abstract:
>
>  *“In contrast, enforcing the inductive biases on a decoder is straightforward”.*
>
> We agree that this writing could give the impression that this is a known fact. Instead, this is something we show in Section 3. We have thus updated this line to:
>
>  *“In contrast, **we show that** enforcing the inductive biases on a decoder is straightforward."*
>
> **Comment**:  “The theoretical claims are based on the assumptions that (1) data is generated by a diffeomorphic function and (2) components in an image can be cut into distinct slots… This limits the use case of the model since real images/other datasets may not satisfy these”
>
> **Response**:
>
> We agree that there may exist certain images in which (1) is not satisfied. For example, for objects with strong occlusions, (1) might not hold since invertibility may break down. Regarding (2), we emphasize that the assumption that the latent space factorizes into slots does not limit the expressivity of the generative model, as the domain of the generator $\boldsymbol{f}$ remains the same irrespective of this assumption.
>
> Furthermore, we note that in our experiments, our learned decoder **does not match the exact form from our theoretical assumptions** but still yields significant gains in OOD performance. This suggests that our assumptions need not be exactly met for models in practice.
>
> Based on the reviewer’s comment, we have added paragraphs titled “Limitations of assumptions in theory” and “Limitations of assumptions in practice” to App. D, discussing our assumptions in more detail.
>
> **Comment**:  “The generative replay is essentially doing synthetic data augmentation. What if, for the feedforward baselines, we also add synthetic data to the training set?”
>
> **Response**:
>
> The reviewer is correct that generative replay adds synthetic data to a training set and that one could in principle add synthetic data to the training set of feedforward baselines to improve OOD performance. Importantly, however, one must first use a generative model to generate this synthetic data. Thus, such an approach necessarily requires leveraging a generator in tandem with a feedforward model making the overall approach generative opposed to non-generative.
>
> **Comment**:  “It is recommended to add pseudo code for the gradient-based search and replay.”
>
> **Response**:
>
> We appreciate the reviewer’s helpful suggestion and have accordingly added pseudocode in Fig. 8, which is referenced in lines 322-323 and 346-347 in the manuscript.
>
> **Comment**:  “Text in Fig 5/6 is too small.”
>
> **Response**:
>
> Based on the reviewers comment, we have increased the text size in both Figures 5 and 6.

---

> > ### Author Response · Authors · 2025-11-21
> > **Official Comment by Authors (2/2)**
> >
> > **Comment**:  “Could the authors elaborate a bit or give an example when the requirements for Eq 2.5/2.6 are not satisfied?”
> >
> > **Response**:
> >
> > Without restrictions on $\mathcal{F}$ the requirements of 2.5/2.6 do not hold in general.
> > To give a concrete example, we consider $\mathcal{Z} = [0,1]^2$ and $\mathcal{Z}_{\textnormal{ID}} = \{(x,y)\in\mathcal{Z} : x = y\}$.
> >
> > Then the functions $\boldsymbol{f}^1(x,y) = 0 \quad \text{and} \quad \boldsymbol{f}^2(x,y) = x - y$ agree in-domain but not out-of-domain.
> >
> > **Comment**:  “The gradient-based search seems to depend on an encoder. How sensitive is it to the errors in the encoder?”
> >
> > **Response**:
> >
> > The reviewer is correct that search depends on the encoder in the sense that the encoder provides the initial starting point for the search procedure (see Fig. 4, left and Fig. 8, top). If we define encoder errors as the OOD performance of the encoder before search/replay, then we do see a trend in Fig. 6B where encoders with lower initial “error” yield higher OOD performance after applying search. However, irrespective of encoder errors, we still see OOD gains for all models after applying search.

---

> > > ### Comment · Reviewer_zHMs · 2025-11-25
> > >
> > > Thanks to the authors for the clarifications. I'll hold the rating due to the theoretical assumptions (or limitations) mentioned in weakness 2.

---

> > > > ### Author Response · Authors · 2025-11-26
> > > >
> > > > Dear reviewer, we are happy that we have addressed your comments such that the only remaining concern regards the limitations of our theoretical assumptions. We respect your concerns to this end. However, we kindly request that you consider two further points when making your decision:
> > > >
> > > > **1.** As previously stated, assumption (2), i.e., the slot assumption, does not reduce the expressivity of the generative process, and thus *does not constitute a meaningful limitation*. Assumption (1), i.e. that $\boldsymbol{f}$ is a diffeomorphism, is standard for theoretical analyses of latent identifiability for image data. Specifically, *every such theoretical study that we are aware of relies on this assumption*. If the reviewer is aware of alternative assumptions, then we are happy to consider them.
> > > >
> > > > **2.** Our goal *is not* to formulate theoretical assumptions that exactly model the natural world. This is generally impossible for any mathematical model. Instead, our goal is to formulate assumptions such that our theory has predictive power in practice. Importantly, we observe exactly this predictive power in our experiments: we do not tailor our data or model to exactly match our theory, yet our empirical results consistently show the limitations of non-generative methods and the success of generative methods, as predicted by our theory. This suggests that our assumptions do not pose limitations on the practical applicability of our theory.
> > > >
> > > > Based on these points, we respectfully request that the reviewer reconsider their decision on raising their score.

---

### Official Review · Reviewer_bY3z · 2025-10-31

**Soundness:** 3
**Presentation:** 3
**Contribution:** 4
**Rating:** 6
**Confidence:** 3

**Summary:**

Generative models have long been proposed to play a critical role in human perception, particularly in explaining feedback connections in the visual cortex. However, the computational advantages of generative models over purely discriminative ones remain debated. This paper provides a theoretical argument that generative models are essential for compositional generalization, as they enable structural constraints to be enforced on learned representations—constraints that cannot be imposed on encoder-only architectures. The paper supports this argument with experiments on the PUG dataset, demonstrating data-efficiency advantages for compositional learning.

The paper is timely, addresses an important question, and proposes a compelling theoretical and conceptual framework. The experimental results are consistent with the theory, although currently limited to a synthetic domain. The work suggests that integrating generative capabilities—or effective generative replay—into representation learning may be necessary to obtain structured inductive biases for compositionality. The contribution would be strengthened by demonstrating these advantages on real visual datasets, but overall, this is a significant and valuable paper.

**Strengths:**

1. Modern computer vision systems are largely discriminative: they encode images for classification or embedding without explicit generative components. In contrast, generative frameworks have been championed in computational neuroscience for explaining feedback and recurrent processing in visual cortex (e.g., Mumford; Lee & Mumford; Rao & Ballard). This paper, therefore, addresses a fundamental and long-standing question: what computational benefits do generative models confer for perception?

2. The work contributes both theoretically and empirically. It articulates why generative models offer advantages—namely, generative architectures can enforce inductive biases and structural constraints that encoder-only systems cannot impose. The experiments using compositional concept/background datasets support this idea by showing that generative models generalize compositionally from limited data, suggesting architectural implications for biological vision systems as well.

**Weaknesses:**

1. All experiments are conducted on the PUG synthetic dataset. While PUG is well-designed for controlled compositional generalization studies, it lacks the semantic richness and visual complexity of natural images (e.g., occlusion, clutter, long-tail distribution shifts). It therefore remains unclear how well the claims would transfer to real-world perception tasks. Testing on real-image compositional benchmarks would strengthen the broader argument that generative mechanisms are required for perception.

2. The results also show that the advantage of generative models diminishes with large-scale training data. Conceptually, this makes sense—large, diverse datasets can approximate compositional coverage, enabling encoder-only models to interpolate rather than extrapolate. However, a deeper explanation or analysis of this transition would improve clarity.

3. Finally, while the proofs appear sound in structure, it is not entirely clear whether the assumptions fully reflect the complexity of natural visual environments. A discussion of the assumptions and their biological and practical implications would be useful.

**Questions:**

What are the limitations of the assumptions made in the mathematical proofs?  Do your assumptions fully reflect the complexity of natural visual environments?  Under what circumstances could your assumptions not appropriate?

---

> ### Author Response · Authors · 2025-11-21
> **Official Comment by Authors**
>
> We thank the reviewer for their time and their comments highlighting the significance of our work. We also appreciate the constructive feedback given by the reviewer. We address each of your comments below:
>
> **Comment**: “All experiments conducted on the PUG dataset which lacks the semantic richness and visual complexity of natural images. Testing on real-image compositional benchmarks would strengthen the broader argument that generative mechanisms are required for perception.”
>
> **Response**:
>
> We fully agree with the reviewer that conducting experiments on datasets of natural images is important. The core challenge is that rigorously evaluating compositional generalization requires datasets in which all ground-truth latent factors in each image (e.g. objects, backgrounds, textures, etc.) are explicitly known, thereby enabling us to construct ID and OOD splits in a principled way. To the best of our knowledge, there currently do not exist “real-image compositional benchmarks” which satisfy this criteria.
>
> Thus, we elected to use PUG, which, to the best of our knowledge, is the most visually complex data in which the underlying latents are known. We fully agree, however, that this dataset has limited complexity relative to natural data. We thus believe that an important direction for future work is to create a large-scale, real-image compositional benchmark where all latents are known. We have now added a paragraph titled “More complex datasets” to App. D of our manuscript discussing these points.
>
> **Comment**: “The results show that the advantage of generative models diminishes with large-scale training data. A deeper explanation or analysis of this transition would improve clarity”
>
> **Response**:
>
> Our current explanation of this transition is that as the pretraining size of the base encoder increases, the full encoder (base encoder plus slot encoder) becomes initialized in a region of the loss landscape where all reachable optima correspond to models that also generalize OOD.
>
> To analyze and understand this transition more deeply, however, would require a more systematic empirical study relative to ours. Namely, one in which an encoder’s architecture and training objective are held fixed, while the model is trained on gradually increasing amounts of pretraining data. We leave such an investigation as an important direction for future work and now include a paragraph titled “Compositional generalization via data scale” in App. D of our manuscript discussing these points.
>
> **Comment**:  “What are the limitations of the assumptions made in the mathematical proofs? Do your assumptions fully reflect the complexity of natural visual environments? Under what circumstances could your assumptions not appropriate?”
>
> **Response**:
>
> Thank you for raising these important questions! Our theory relies on two main assumptions on the ground-truth generator $\boldsymbol{f}$: (i) that $\boldsymbol{f}$ is a diffeomorphism and (ii) that the interactions between slots under $\boldsymbol{f}$ are restricted according to Eq. 2.7. While these assumptions are the most general which have been shown to enable compositional generalization, it is possible that they may fail to capture real-world images in certain contexts. For example, the assumption that $\boldsymbol{f}$ is a diffeomorphism may not hold for images in which objects exhibit strong occlusions such that $\boldsymbol{f}$ is no longer invertible. Additionally, it is possible that certain complex interactions between concepts may not be captured by 2.7 such as complex shadows or reflections.
>
> **Comment**: “A discussion of the assumptions and their biological and practical implications would be useful.”
>
> **Response**:
>
> Based on the reviewer's comment we have added a paragraph in App. D: “Limitations of assumptions in theory” discussing our theoretical assumptions and their limitations. We have also included an additional paragraph “Limitations of assumptions in practice” discussing the practical implications of these assumptions as well as a brief note on biological plausibility.

---

### Official Review · Reviewer_keEf · 2025-10-31

**Soundness:** 3
**Presentation:** 3
**Contribution:** 3
**Rating:** 6
**Confidence:** 4

**Summary:**

The paper presents a theoretical framework and empirical evidence supporting the necessity of generative models for achieving data-efficient compositional generalization in visual perception tasks. The core argument is that generative inversion (i.e., using a decoder to reconstruct data from latent representations) is fundamentally required to guarantee generalization to out-of-distribution (OOD) data due to the nature of the inverse problem on high-dimensional data manifolds. This is contrasted with non-generative (encoder-based) approaches, which are shown to struggle with OOD generalization unless massively scaled. Experimentally, they demonstrate that Non-generative models (encoders trained with supervised or unsupervised objectives) fail at OOD compositional generalization when trained from scratch or with limited pretraining. Performance only emerges with massive scale (e.g., SigLIP2). WhileGenerative models (autoencoders), when combined with inversion techniques like gradient-based search (online optimization) and generative replay (offline recombination of learned slots), achieve significantly higher OOD accuracy across all datasets.

**Strengths:**

1. Strong Theoretical Foundation: Provides rigorous proofs establishing the infeasibility of enforcing compositional generalization guarantees on encoders (Theorem 3.2, Lemma 3.1, A.4) and the feasibility for decoders (Section 3, Theorem A.8).
2. Compelling Empirical Validation: Uses controlled, photorealistic datasets (PUG) to demonstrate the practical limitations of non-generative models (Fig 5) and the practical benefits of generative approaches + inversion techniques (Fig 6).
3. Novel Insights on Inversion Techniques: Clearly demonstrates the power of combining generative pretraining (autoencoders) with efficient inversion methods (gradient search, generative replay) for OOD generalization, going beyond standard VAE training.
4. Clarity of Core Argument: The central thesis – that generation is required for data-efficient compositional generalization – is well-motivated, rigorously supported, and clearly articulated,  both in theory and empirical practice.

**Weaknesses:**

1. Computational Cost: The generative + search/replay approach is inherently more computationally expensive (per-query optimization or large generative model) than a single forward pass through an encoder. This practical trade-off is acknowledged but not deeply analyzed.
2. Scalability to Complex Real-World Data: While PUG is controlled, experiments don't scale to the complexity of full ImageNet or real-world uncurated data.
3. Limited Exploration of Alternative Generative Setups: Primarily uses a specific autoencoder architecture. The performance gains are attributed to the generative paradigm + inversion, but the impact of specific architectural choices (beyond the regularization from Brady et al.) could be explored more.
4. Related Work Integration: While related work is discussed, existing works on diffusion/generative classifiers could potentially be integrated more deeply into the narrative or limitations discussion.

**Questions:**

See the weakness.

---

> ### Author Response · Authors · 2025-11-21
> **Official Comment by Authors**
>
> We thank the reviewer for their time and their positive assessment of both our theoretical and empirical contributions. We also appreciate the detailed feedback given by the reviewer. We address each of your comments below:
>
> **Comment**: “The generative + search/replay … is more computationally expensive than a single forward pass through an encoder. This practical trade-off is … not deeply analyzed.”
>
> **Response**:
>
> We agree with the reviewer that a more in depth analysis of the computational overhead of generative methods is important to include. To this end, we have added a paragraph titled “Computational cost of generative methods” to App. D which addresses this point.
>
> **Comment**: “While PUG is controlled, experiments don't scale to the complexity of full ImageNet or real-world uncurated data.”
>
> **Response**:
>
> We fully agree that conducting experiments on datasets with the complexity of real-world data is important. The core challenge is that rigorously evaluating compositional generalization requires datasets in which all ground-truth latent factors in each image (e.g. objects, backgrounds, textures, etc.) are explicitly known, thereby enabling us to construct ID and OOD splits in a principled way. For real-world, unstructured datasets such as ImageNet, the underlying latents are unknown making such datasets unsuitable for testing compositional generalization.
>
> Consequently, we elected to use PUG, which to the best of our knowledge is the most visually complex data where the underlying latents are known. Nevertheless, the dataset has limitations in its visual complexity. Thus, we believe an important direction for future work is to create a large-scale real-world dataset with access to ground-truth latents, perhaps by leveraging synthetic or neural network based image renders. We have added a paragraph titled “More complex datasets” to App. D discussing these points.
>
> **Comment**:  “Primarily uses a specific autoencoder architecture. … the impact of specific architectural choices could be explored more.”
>
> **Response**:
>
> We agree with the reviewer that testing different generative setups is an important direction to explore. To this end, we have conducted experiments using an unstructured decoder which does not satisfy our assumptions. We discuss these results further in our general reply and in detail in Appendix C which we reference in the main text in lines 409-410.
>
> To briefly summarize, however, we find that this unstructured decoder yields significantly lower OOD performance than our structured Transformer decoder highlighting the importance of inductive bias in achieving compositional generalization for generative methods as suggested by our theory.
>
> **Comment**:  “existing works on diffusion/generative classifiers could potentially be integrated more deeply”
>
> **Response**:
>
> Based on the reviewers comment, we have integrated these related works into lines 1410-1412 in the limitations paragraph “Limitations of assumptions in practice”.

---

> > ### Comment · Reviewer_keEf · 2025-11-25
> > **Post-rebuttal comment**
> >
> > Thanks for the detailed response from the authors. Most of my concerns have been addressed and I decide to maintain my initial score as 6 due to the limited evaluation of more complex uncurated datasets somewhat weakens its effectiveness and generalization across real-world applications.

---

> > > ### Author Response · Authors · 2025-11-27
> > >
> > > Dear reviewer,
> > >
> > > Thank you for your response. We are pleased that most of your concerns have been addressed.
> > >
> > > We would like to clarify that evaluating compositional generalization on “uncurated datasets” is *not possible*, but requires curated ID and OOD splits. In our experiments, we use what is, to the best of our knowledge, the *most complex* image dataset that provides these curated splits. If you are aware of an alternative dataset that satisfies these criteria, we would be very happy to consider it.
> > >
> > > We would also like to emphasize that non-generative methods already fail on our dataset, which is less complex than real-world images. It is therefore unlikely that these methods would perform better as image complexity increases and the task becomes harder. In this sense, our experiments already provide a meaningful signal about performance on real-world images.
> > >
> > > In light of these points, we respectfully request that you reconsider your decision on raising your score.

---

### Official Review · Reviewer_dnFU · 2025-11-01

**Soundness:** 2
**Presentation:** 3
**Contribution:** 2
**Rating:** 4
**Confidence:** 3

**Summary:**

This paper investigates whether non-generative and generative perception approaches can generalize to out-of-distribution data by simply enforcing constraints on architecture both theoretically and empirically. They show that it is infeasible to simply constrain the structure of encoder to achieve compositional perception, and optimization process is the key for non-generative approaches to generalize OOD. In contrast, it is easy to enforce inductive bias on decoder to achieve compositional generalization. Empirically, multiple non-generative and generative approaches are evaluated in terms of generalization performance.

**Strengths:**

1. The paper is well-written and easy to follow.

2. The paper is well motivated. There has been extensive discussions in the literature regarding generative and non-generative perception approaches for a long time, including many empirical studies. However, theoretical analysis regarding this is still missed. This work is trying to bridge this gap.

**Weaknesses:**

1.  Most of theoretical results are based on [Brady et al,. 2025], and thus obtaining results of Lemma 3.1 and Theorem 3.2. seem a bit straightforward.

2. For gradient-based search, when using encoder to provide initial guess, it is hard to determine if part of the compositionality performance of decoder comes from encoder. For Figure 6 B, is an additional encoder used?

3. For generative replay, though not explicitly requiring external data, it is essentially using learned generative model to augment the dataset, such that OOD data are included in IID data when training an additional encoder. It hinges on the generative performance of a decoder. Meanwhile, can we say that the compositionality is traded with extra computation and model?

4. The paper emphasizes that compositional performance of non-generative approaches is only possible by using optimization techniques. However, telling from Figure 6, it looks like generative approaches rely more on optimization strategies (e.g., encoder results as initialization, replay).

5. Figure 6 says reporting OOD performance across three datasets, but actually shows only two. Any reason to drop PUG-object?

6. The paper claims that generative models for perception are more data-efficient than non-generative approaches. However, evidence to support this is not so clear. Especially, the design of experiments is very entangled in the sense that non-generative encode and generative decoder are trained jointly. It is hard to determine if the compositional perception performance of decoder also comes from using features from pretained model in the pipeline. Also, given small dataset, is it possible that encoder (large) overfits while decoder (relatively small) not? A further question is why not training both encoder and decoder with small architecture from scratch on your small dataset for comparison?

**Questions:**

In 3.1, it is said that inverse generators g do not admit an analytical form similar to Eq. (2.7). Is this always the case? or it is a general assumption to facilitate theoretical analysis? Is it possible to make it this form in practice?

---

> ### Author Response · Authors · 2025-11-21
> **Official Comment by Authors (1/2)**
>
> We thank the reviewer for their time and appreciate their detailed feedback on our work. We address each of your comments below:
>
> **Comment**: “Most theoretical results are based on [Brady et al,. 2025], thus obtaining results … seems straightforward”
>
> **Response**:
>
> The reviewer is correct that many of our assumptions are based on [Brady et al,. 2025]. We respectfully disagree, however, that our results follow in a straightforward manner from this work.
>
> [Brady et al,. 2025] studied functions of the form in Eq. 2.7. Conversely, our theoretical results study the inverses of such functions $\mathcal{G}\_\textnormal{int}$. Functions $\mathcal{G}\_\textnormal{int}$ do not offer an analytical form. Thus, understanding their structure is not straightforward and requires a detailed theoretical analysis. To this end, we:
>
> 1. Prove that the Hessian of $\mathcal{G}\_\textnormal{int}$ can have arbitrary structure in cases when $d_x \gg d_z$ (Theorem 3.2)
> 2. Derive the tangent space structure of $\mathcal{G}\_\textnormal{int}$  when $d_x \gg d_z$ (Eq. 3.4, Lemma A.4).
> 3. Conduct a theoretical analysis of whether architectural inductive biases can be used to enforce that an encoder $g \in \mathcal{G}\_\textnormal{int}$ (App. A.2).
>
> None of these results are trivial to show nor follow as corollaries from [Brady et al,. 2025]. We thus believe our theoretical contribution offers substantial novelty relative to this prior work.
>
> **Comment**: “why not training both encoder and decoder with small architecture from scratch on your small dataset for comparison”
>
> **Response**:
>
> We note that we do in fact train a small model from scratch on our datasets which is stated in line 402 in the original manuscript. This model uses a ViT-Small for the base encoder and is labelled in Figs 5 and 6 as “From scratch (ViT-S/16)”.
>
> **Comment**: “It is hard to determine if the compositional performance of decoder also comes from using features from pretrained model”
>
> **Response**:
>
> As previously noted, we include a model trained from scratch in Figs 6A, 6B which does not rely on a pre-trained encoder. This model yields substantial gains in OOD performance when leveraging a decoder with replay and search. This indicates that decoder compositionality does not require leveraging a pre-trained encoder.
>
> **Comment**: “For gradient-based search … it is hard to determine if … the compositionality … of decoder comes from encoder. For Figure 6 B, is an additional encoder used?”
>
> **Response**:
>
> To answer the reviewers comment, we first briefly recap our experimental setup in Fig. 6B:
>
> We first train an autoencoder which consists of an encoder and a decoder. We then evaluate OOD classification accuracy using representations given by the trained encoder. These scores are the blue bar in Fig. 6B and can be understood as compositionality/OOD performance coming solely from the encoder. We then refine the encoder’s representation via gradient based search using our trained decoder and compute the OOD accuracy of the resulting representation (yellow bar).
>
> During this search procedure, the encoder’s weights are frozen and only the latent representation is updated. Thus, these gains in compositionality/OOD performance can be attributed solely to the decoder. This indicates that the decoder offers gains in compositionality which are independent of the encoder. Further, an additional encoder is not used to give the initial representation for search but instead the same encoder used during training.
>
> **Comment**: “is it possible that encoder (large) overfits while decoder (relatively small) not?”
>
> **Response**:
>
> In short, we do not believe it is the case that the encoder is overfitting while the decoder is not. In our experiments in Fig 6, the bulk of the parameters of the encoder, i.e. those in the pre-trained ViT, are frozen, and only the smaller “slot-encoder” is trained. Thus, there is not a significant discrepancy in parameter count between the trained encoder and decoder.
>
> Furthermore, the encoder and decoder are trained to be inverses of each other on the training data. Thus, it is unclear how a situation could arise in which the encoder overfits the training data but the decoder does not.
>
> **Comment**: “The paper claims that generative models for perception are more data-efficient than non-generative approaches. However, evidence to support this is not so clear..”
>
> **Response**:
>
> We hope that our responses above have provided adequate evidence to address this concern. If any doubts still remain to this end, then we are happy to provide further clarification!

---

> > ### Author Response · Authors · 2025-11-21
> > **Official Comment by Authors (2/2)**
> >
> > **Comment**: “For generative replay … can we say that the compositionality is traded with extra computation and model?”
> >
> > **Response**:
> >
> > The reviewer is correct that achieving compositionality through search or replay using a decoder introduces a tradeoff between compositionality and computation, and we have now included a paragraph “Computational cost of generative methods.” in App. D discussing this computational overhead.
> >
> > We also note that being able to trade compositionality for compute can, in some cases, be seen as an advantage of our framework since this offers explicit control over the amount of computation allocated to each input. This is analogous to modern LLMs, where one may choose between fast “System 1” responses and more refined, but compute intensive “System 2”, reasoning-based responses. Our framework offers analogous control over this System 1, System 2 tradeoff through increasing the number of search or replay iterations.
> >
> > **Comment**: “The paper emphasizes that compositional performance of non-generative approaches is only possible by using optimization techniques. However,..., it looks like generative approaches rely more on optimization”
> >
> > **Response**:
> >
> > We believe the reviewer’s interpretation of “depend on the optimizer” (line 284 in our original manuscript) differs from our intended meaning. We apologize for this confusion and aim to clarify below.
> >
> > Our point is not that non-generative methods rely on optimization for compositional generalization while generative methods do not. As the reviewer correctly notes, both methods involve optimization.
> >
> >
> > The distinction we aim to highlight is this:
> > Our theoretical results in Section 3 show that generative methods can enforce the inductive biases required for compositional generalization, whereas non-generative methods cannot. As a consequence, every minimizer of the loss for generative methods yields a solution that generalizes compositionally. In contrast, for non-generative methods, there exist models that minimize the loss but fail to generalize. Thus, for non-generative methods, whether a model achieves compositional generalization “depends” on whether the optimizer happens to arrive at a solution which generalizes compositionally.
> >
> > We have rewritten the “Takeaways” paragraph at the end of Section 3 to communicate this point more clearly.
> >
> > **Comment**: “Figure 6 says reporting OOD performance across three datasets, but actually shows only two. Any reason to drop PUG-object?”
> >
> > **Response**:
> >
> > We appreciate the reviewer pointing out this typo! Fig. 6 does indeed only include results for two datasets, i.e., PUG-Background and PUG-Texture. The reason for not including PUG-Object is that it is the one dataset in which non-generative methods are able to achieve near perfect OOD accuracy. Consequently, applying search or replay to the encoders from this dataset does not make sense, since further gains in OOD performance are not possible.
> >
> > We have added a sentence explaining this point (lines 438-440) along with correcting the typo in the caption for Fig. 6.
> >
> > **Comment**: “it is said that inverse generators g do not admit an analytical form. Is this always the case? or it is a general assumption to facilitate theoretical analysis? Is it possible to make it this form in practice?”
> >
> > **Response**:
> >
> > We emphasize that we do not assume that inverse generators do not have an analytical form. Instead, we assume that generators are of the form in Eq. 2.7. We then study the inverses of such functions and highlight that such functions do not admit analytic structure nor do they admit special derivative structure similar to their inverses.

---

> > > ### Comment · Reviewer_dnFU · 2025-11-27
> > >
> > > Thank you for the rebuttal. Most of my concerns are addressed. I raise the rating to 6.

---

> > > > ### Author Response · Authors · 2025-11-28
> > > >
> > > > We are pleased to hear that most of your concerns have been addressed. Thank you for your feedback and time.

---

### Author Response · Authors · 2025-11-21
**General Reply to All Reviewers**

We thank the reviewers for their valuable feedback, and their assessment of our work as “significant and valuable“ and “addressing a fundamental and longstanding question” `bY3z`. Furthermore, we appreciate that the reviewers found our theoretical contribution “strong” and “rigorous” `keEf`, aspects of our proposed method as “novel” `zHMs`, and our core argument as “well-motivated and rigorously supported … both in theory and practice” `keEf`.

We are grateful for the constructive feedback given by all reviewers. Based on this feedback, we have made several additions to the manuscript which are highlighted in yellow in the revised version. We briefly summarize the most significant additions below:

**Experiments with Different Decoders**

Reviewer `keEf` noted the importance of exploring different generative setups in our experiments. To address this, we have added additional experiments to Appendix C, which compare the compositional generalization of an unstructured CNN decoder which does not meet our theoretical assumptions, vs. the structured Transformer decoder used in our experiments, based on our theory. We find that the unstructured CNN decoder yields a nearly 40% drop in OOD accuracy relative to the structured decoder (Fig. 9), highlighting the importance of placing inductive biases on a decoder as suggested by our theory.

**Additional Discussion Sections**

Based on the reviewer's comments, we have added a new section (Appendix D) which includes several additional discussion paragraphs. Namely, we have added discussions on:
* Conducting experiments on real-world image data
* Limitations of our assumptions in theory and practice
* Better understanding how data scale can yield compositional generalization

We go into detail on these discussions in our individual replies.

**Clarity of Manuscript**

Reviewer `zHMs` raised several points regarding the clarity of our manuscript when discussing the distinction between generative vs. non-generative methods. Based on this feedback, we have re-written several sentences in the Abstract, Introduction, and Section 2, to improve clarity. Reviewer `dnFU` also highlighted ambiguous writing in Section 3, leading us to rewrite the “Takeaways” paragraph at the end of this section. We discuss these points further in our individual replies.

---

### Meta-Review · Area_Chair_ZpAN · 2025-12-27

**Summary:**

While the paper addresses an important and long-standing problem and proposes a theoretically sound framework, its practical impact is limited. The empirical evaluation is confined to a controlled synthetic benchmark (PUG), raising concerns about scalability and relevance to real-world settings. The proposed generative inversion mechanism introduces additional computational overhead, and it remains unclear whether the reported gains stem from the core theoretical ideas or from confounding factors such as architectural choices, optimization effects, or implicit data augmentation. Moreover, the theoretical assumptions, although standard, may be overly restrictive in practice. As a result, despite strong conceptual motivation, the work falls short in demonstrating sufficient empirical breadth and real-world applicability.

**Reviewer Concerns:**

**Addressed Concerns**

- Clarity of generative vs. non-generative distinction (Reviewer zHMs, dnFU):
The authors substantially revised the abstract, introduction, and Section 2 to clarify definitions (image encoder/decoder vs. transformer encoder/decoder), resolving confusion.
- Dependence on prior theory (Brady et al.) (Reviewer dnFU): The rebuttal convincingly clarified novelty by emphasizing analysis of inverse generators, Hessian structure, and encoder limitations that do not follow directly from prior work.
- Role of encoder vs. decoder in search-based gains (Reviewer dnFU, zHMs): The authors clarified that encoder weights are frozen during search and that gains arise from decoder inversion, resolving attribution concerns.
- Computational cost and trade-offs (Reviewers keEf, dnFU): Explicit discussion was added (Appendix D), acknowledging compute–performance trade-offs and framing them as controllable (System 1 vs. System 2 analogy).
- Limited exploration of generative architectures (Reviewer keEf): Additional experiments with an unstructured decoder were added, empirically supporting the theoretical claim about decoder inductive bias.
- Missing or unclear experimental details (figures, typos, pseudocode) (Reviewers dnFU, zHMs): Errors were corrected, figures clarified, and pseudocode for search/replay was added.

**Outstanding Concerns**

- Generality beyond synthetic datasets (Reviewers keEf, bY3z, zHMs): Despite justification, reviewers remain unconvinced that results transfer to real-world visual complexity due to exclusive reliance on PUG.
- Strength of theoretical assumptions (Reviewers bY3z, zHMs): Assumptions such as diffeomorphism and slot factorization, while standard, are still seen as limiting, particularly for natural images with occlusion and complex interactions.
- Diminishing advantage at scale (Reviewer bY3z): The observation that large-scale pretraining reduces the generative advantage is acknowledged but not deeply analyzed.

These remaining concerns primarily affect scope and generalization, rather than correctness or internal consistency.

**Reviewer Scores:**

- **Reviewer dnFU**: Score increased from 4 → 6, as most theoretical, experimental, and clarity concerns were satisfactorily addressed in the rebuttal.
- **Reviewer keEf**: Score remained at 6, maintaining a borderline-positive stance due to unresolved concerns about evaluation on complex real-world data.
- **Reviewer bY3z**: Score remained at 6, continuing to view the work as significant and valuable, but limited by synthetic-only validation.
- **Reviewer zHMs**: Score remained at 6, with the primary remaining concern being the strength and realism of theoretical assumptions.

---

### Decision · Program_Chairs · 2026-01-26

Reject